# Modeling of Reactive Sputtering—History and Development

**DOI:** 10.3390/ma16083258

**Published:** 2023-04-20

**Authors:** Viktor I. Shapovalov

**Affiliations:** Department of Physical Electronics and Technology, St. Petersburg Electrotechnical University “LETI”, Prof. Popov Str., 5F, 197022 St. Petersburg, Russia; vishapovalov@mail.ru

**Keywords:** reactive sputtering, model, chemisorption, implantation, knock-on effect, chemical reaction, law of mass action, Langmuir isotherm

## Abstract

This work critically reviews the evolution of reactive sputtering modeling that has taken place over the last 50 years. The review summarizes the main features of the deposition of simple metal compound films (nitrides, oxides, oxynitrides, carbides, etc.) that were experimentally found by different researchers. The above features include significant non-linearity and hysteresis. At the beginning of the 1970s, specific chemisorption models were proposed. These models were based on the assumption that a compound film was formed on the target due to chemisorption. Their development led to the appearance of the general isothermal chemisorption model, which was supplemented by the processes on the surfaces of the vacuum chamber wall and the substrate. The model has undergone numerous transformations for application to various problems of reactive sputtering. At the next step in the development of modeling, the reactive sputtering deposition (RSD) model was proposed, which was based on the implantation of reactive gas molecules into the target, bulk chemical reaction, chemisorption, and the “knock-on effect”. Another direction of the modeling development is represented by the nonisothermal physicochemical model, in which the Langmuir isotherm and the law of mass action are used. Various modifications of this model allowed describing reactive sputtering processes in more complex cases when the sputtering unit included a hot target or a sandwich one.

## 1. Introduction

Films of simple compounds of metals (oxides, nitrides, carbides, oxynitrides, etc.) and their solid solutions are of great interest in many fields of technology. This is due to the fact that they exhibit many properties (semiconductor, ferromagnetic, ferroelectric, electrochromic, photochromic, etc.) that open up new opportunities for innovation. The attention of scientists and engineers to such films is also associated with new areas of their application, which include ecology, medicine, and alternative energy.

For the deposition of these films, there are widely used methods summarized by the term “reactive sputtering” [1,2], which, according to the author [2], first appeared in [3]. Let us denote here the metal, the reactive gas, and their compound with stoichiometric coefficients *m* and *n* by M, X_2,_ and M*_m_*X*_n_*, respectively. Regardless of the sputtering system type, several methods are applied to deposit M*_m_*X*_n_* films [4]:Reactive sputtering of the M-target in the Ar + X_2_ environment. The method produces a film of stoichiometric composition;Sputtering of the M*_m_*X*_n_* target in the Ar + X_2_ environment. The method also leads to a stoichiometric film;Sputtering of the M*_m_*X*_n_* target in the Ar environment. Due to the preferential sputtering of the light component X_2_, the surface layer of the target is reduced from M*_m_*X*_n_* to MX*_x_*, which allows the depositing of films of intermediate composition.

In early experimental works, the influence of the reactive gas concentration in the gas mixture or its partial pressure on the film growth rate, discharge voltage, and film composition was discovered. Such results were obtained for oxides ZrO_2_ [5], ZnO [6], SiO_2_ [7], TiO_2_ [8], TiN nitrides [9], and other materials [10,11,12,13,14,15,16,17,18].

In addition, during the deposition of films of oxides TiO_2_ [10,13,14], Fe_2_O_3_, CoO, and Ag_2_O [15] and nitrides TiN [19,20] and ZrN [21], nonlinear effects were found, and it was established that there is a critical value with an increase in the partial pressure of the reactive gas. At this value, during the sputtering process, an abrupt change in the film deposition rate, in some cases by an order of magnitude, was observed [15].

In later works, it was found that the partial pressure of the reactive gas is not an independent variable. It reflects only the state of the reactive sputtering process at given values of other parameters that could be changed independently. The main ones were the incoming reactive gas flow *Q*_0_ and the discharge current *I* (or the power released on the target). In studies at *I* = const and *Q*_0_ = var, an avalanche-like change in the partial pressure of the reactive gas was observed [14]. A similar effect was observed at *Q*_0_ = const and *I* = var [22]. However, in this case, an avalanche-like change in the partial pressure was observed at a certain value of the discharge current (or power). Such results have been obtained for many compounds. In particular, they include TiO_2_ [23], In_2_O_3_ [10], Ta_2_O_5_ [11], SiO_2_ [18], TiN [20], etc.

Considering the process of reactive sputtering in general terms, two steady modes of operation of the target were distinguished: metallic and reactive [24].

In the metallic mode:The target is free from reaction products;Metal is sputtered from the target surface. The sputtering rate is high, and the ion-induced electron emission yield is not high;The reactive gas adsorbs on the target, the inner surfaces of the vacuum chamber, on which atoms of the sputtered metal are deposited, and is pumped out;Current–voltage (I–V) characteristic of the discharge corresponds to the discharge in pure argon.

In the reactive mode:The target surface is completely covered by the M*_m_*X*_n_* compound;M*_m_*X*_n_* molecules are sputtered from the target surface. The sputtering rate is low, and the ion-induced electron emission yield is high;The target consumes an insignificant part of the reactive gas to maintain the steady state, the rest of the gas is pumped out by a vacuum pump;The I-V characteristics of the discharge differ from that of the metallic mode—the direction and magnitude of the changes depend on the material of a target and the type of reactive gas.

Getting to the point of instability, the process of reactive sputtering spontaneously passes into a new steady state. In a large number of experiments, when changing *Q*_0_ or the discharge current (power), the hysteresis effect was found. It was observed during the deposition of oxide films TiO_2_ [25,26], Al_2_O_3_ [27,28], Cd_2_SnO_4_ [29,30], ZnO [31], Ta_2_O_5_ [32], Ta_2_O_5_–TiO_2_ [33] and nitrides AlN [34] and TiN [22,24,35]. The essence of the hysteresis resides in the following: there are two points of instability in the dependence of the X_2_ reactive gas partial pressure on *Q*_0_ at *I* = const (Figure 1a, curve *1*). With an increase in *Q*_0_ from zero, the target remains in the metallic mode up to point A. With this value of *Q*_0_, the target avalanches into the reactive mode. A return to the metallic mode with a decrease in *Q*_0_ occurs at another point C. Curve *2* in Figure 1a reflects the corresponding dependence when the magnetron is turned off.

A similar nonlinearity was found in processes in which the discharge current is controlled at *Q*_0_ = const (Figure 1b). In this case, as *I* increases from zero, the target remains in the reactive mode up to point A. At this value of the current, the target avalanches into the metallic mode. A return to the reactive mode with a decrease in *I* occurs at another point, C.

Thus, numerous experiments have shown, firstly, that the main independent variables of the reactive sputtering process are the reactive gas flow *Q*_0_ and the discharge current *I* (the power released on the target). Secondly, the curves of the dependence of the partial pressure of the reactive gas *p* on *Q*_0_ (Figure 1a) at *I* = const contain several sections:In the initial section 0A on curve *1*, the value of *p* is close to zero (the total pressure of the gas mixture is equal to the partial pressure of argon);As *Q*_0_ increases to point A, a noticeable change in *p* is not observed;When point A is reached, the process avalanches into a new steady state (point B);With a further increase in *Q*_0_, the pressure *p* grows in proportion to *Q*_0_, but for any *Q*_0_ it is always less than the value measured in the absence of a discharge (curve *2* in Figure 1a);As *Q*_0_ decreases to point C, a proportional decrease in *p* is observed;When point C is reached, the process returns like an avalanche to the initial steady state (point D).

When controlling the process of reactive sputtering by incoming flow *Q*_0_ or discharge current *I*, it is possible to deposit the M*_m_*X*_n_* film only in the reactive mode. The points in Figure 1a, corresponding to this mode, are located on the line connecting points B and C. In Figure 1b, similar points are on the line connecting points A and D. Points A and C in Figure 1 are connected by a dashed line, which can only be reached by the sputtering process under specially created conditions.

These conditions are described in [20,36]. The hysteresis in them was overcome by changing the controlled variable and introducing negative feedback. When sputtering a titanium target in the Ar + N_2_ environment, instead of controlling the N_2_ incoming flow, an adjustable partial pressure system was used. As a result, continuous dependencies were obtained that relate the partial pressure of N_2_ to its incoming flow. The curves were N-shaped with areas of negative slope. This area was not available when controlling the incoming flow of nitrogen.

Similar dependences were obtained by sputtering an aluminum target in the Ar + N_2_ environment with discharge voltage control [34,37]. The presented results show that in systems with negative feedback on reactive gas pressure or discharge voltage, the sputtering process can be maintained at any point of the sections AC in Figure 1.

A brief review of the experimental results, indicating the features of reactive sputtering, showed the complexity of these processes. Therefore, it was difficult to choose the film deposition mode, especially at the initial stage of technology development. For a more detailed study of the processes of reactive sputtering, many researchers worked on the creation of their models.

Today it is obvious that the key processes in reactive sputtering occur on the target. On its surface, the processes of formation and sputtering of the M*_m_*X*_n_* film compete, and depending on the ratio of their rates, the target, as indicated earlier, can be in one of two steady states. The state of the target is quantitatively described using the relative fraction of its surface θ_t_ covered by the M*_m_*X*_n_* film.

In the most general form, the kinetic equation for the target can be written as:$$\frac{d\theta_{t}}{dt}\;\;=\;\;\sum_{i}{\left(\frac{d\theta_{t}}{dt}\right)}_{i}^{+}+\;\sum_{j}{\left(\frac{d\theta_{t}}{dt}\right)}_{j}^{-},$$

With the help of the first term on the right side of the written equation, a set of processes that increase the value of θ_t_ is taken into account. The second term describes the decrease in θ_t_.

A description of the history and development of these works is the purpose of this article. Let us first consider the early models, which we will call “specific” (Section 2). Next, let us turn our attention to publications presenting the Berg isothermal chemisorption models (Section 3), the similar Depla models (Section 4), and the Barybin nonisothermal physicochemical models (Section 5).

## 2. Specific Isothermal Models

All the varieties of known models of the reactive sputtering process are actually based on two assumptions:Two processes compete on the excited target surface: the formation of a thin layer of metal compound with reactive gas and sputtering of this layer by accelerated argon ions;On the substrate and walls of the vacuum chamber, the sputtering target material is deposited, and the reactive gas molecules are chemisorbed.

The physical model of processes on the target surface was first discussed in 1973 [15]. In 1975 [21,38], kinetic equations for the target based on chemisorption appeared.

In early works [19,21,22,26,29,37,39,40,41], when constructing a reactive sputtering model, the partial pressure was taken as an independent variable *p* of the reactive gas and took into account only the processes occurring on the target without taking into account its temperature. The difference between the models of different authors was not fundamental. The main unifying element of all the noted publications was the kinetic equation for the target surface, which in the steady mode took the form:(1)$$\frac{dN}{dt}=\alpha \frac{p_{\mathrm{eff}}}{\sqrt{2\pi m_{0}kT}}\left(1-\theta_{t}\right)-\frac{S_{C}\left(E\right)j}{e}\theta_{t}=0,$$
where *N* is the density of adsorption centers occupied by the reactive gas; *α* is the coefficient of adsorption of reactive gas molecules to the metal surface of the target; *p*_eff_ is the effective pressure during sputtering; *m*_0_ is the mass of a gas molecule; *k* is the Boltzmann constant; *T* is the absolute temperature; θ_t_ = *N*/*N*_t_ is the fraction of the target surface covered by the compound film; *N*_t_ is the density of adsorption centers on the target; *S*_C_ is the M*_m_*X*_n_* film sputtering yield; *j* is the discharge current density on the target. The effective pressure in (1) was defined as:(2)$$p_{\mathrm{eff}}=p-\frac{a}{S_{p}}A_{t}\frac{S_{M}j}{e}\left(1-\theta_{t}\right),$$
where *a* is coefficient converting the number of sputtered atoms into the rate of gas adsorption; *S*_p_ is the pumping speed of the chamber; *A*_t_ is the target area; *S*_M_ is the metal sputtering yield.

The first term in (1) determines the growth rate of the M*_m_*X*_n_* compound, the factor $p_{\mathrm{eff}}/\sqrt{2\pi m_{0}kT}$, according to the kinetic theory of gases, sets the density of the reactive gas molecules flowing to the target. The second term determines the sputtering rate of the target surface covered by the M*_m_*X*_n_* film by argon ions. The second term in expression (2) takes into account the decrease in pressure due to the adsorption of the reactive gas on the chamber walls.

Solution (1) with respect to (1 − θ_t_) has the form:$$\left(1-\theta_{t}\right)=\left(B-\sqrt{B^{2}-4AC}\right){\left(2A\right)}^{-1},$$
where *A*, *B,* and *C* are the coefficients depending on the sputtering process parameters.

In [29], Equation (1) was written by its authors using the M*_m_*X*_n_* film thickness on the *x* target surface. The unbalance in the sputtering system, at which the proportion of the target surface occupied by the compound begins to increase, is described by the equation
$$\frac{dx}{dt}={\left(\frac{dx}{dt}\right)}_{\mathrm{ch}}+{\left(\frac{dx}{dt}\right)}_{\mathrm{sp}},$$
where (*dx*/*dt*)_ch_ is the compound film formation rate; (*dx*/*dt*)_sp_ is the sputtering rate of the M*_m_*X*_n_* layer. According to [15], the film formation rate is expressed in exponential form:$${\left(\frac{dx}{dt}\right)}_{\mathrm{ch}}=\frac{A(p)}{\rho }\exp\left(-\frac{x}{x_{0}}\right),$$
where *A*(*p*) is an increasing function of the reactive gas pressure; ρ is the compound density; *x*_0_ is a temperature-dependent parameter. The change in the partial pressure of oxygen upon unbalance is defined as:$$p\left(t\right)=A-B\frac{dx}{dt},$$
where *A* and *B* are the constants, depending on system parameters.

In [37], for the development of the model described by Equation (1), two mechanisms were proposed that form a compound of metal and reactive gas atoms on the target surface:Chemisorption of neutral reactive gas molecules (which can occur without a glow discharge);Covering the target with ions and atoms of reactive gas, which are activated by a glow discharge. This process is called by the authors “ion plating”.

Kinetic Equation (1), in this case, took the form
(3)$$\frac{dN}{dt}=\alpha F+f\left(\frac{p}{p_{t}}\right)\alpha_{i}\left(\theta_{t}\right)\frac{I}{\left(1+\gamma \left(\theta_{t}\right)\right)e}-\frac{S_{C}\left(\theta_{t}\right)I}{\left(1+\gamma \left(\theta_{t}\right)\right)e},$$
where *F* is the density of the flow of reactive gas molecules to the target, which is defined by the Hertz–Knudsen equation; *f*(*p*/*p_t_*) is the proportion of reactive gas positive ions in the discharge current; *p_t_* is the total pressure. The rest of the parameters have different values on the clean and coated parts of the M*_m_*X*_n_* target. In this regard, their values, when averaged over the entire surface of the target, will depend on θ_t_; α*_i_*(θ_t_) is the adsorption coefficient for reactive gas ions incident on the target surface; γ(θ_t_) is the average ion-induced secondary electron emission yield for the total target surface; *S*_C_(θ_t_) is the average sputtering yield of reactive gas molecules on the target.

The first term on the right side of Equation (3) takes into account the chemisorption of neutral reactive gas molecules on the target surface, the second term—ion deposition associated with the discharge current, and the third term—the sputtering of reactive gas molecules from the target surface. The considered model, according to the authors, allows for determining the composition of the compound film from the intensity of the metal line in the discharge emission spectrum and other process parameters.

Summing up, we note that all models of reactive sputtering, which are described by Equation (1), combine several assumptions:The partial pressure of the reactive gas and the film deposition rate are determined by the degree of coverage of the target surface with the M*_m_*X*_n_* film;Chemisorption was adopted as the mechanism of the M*_m_*X*_n_* film formation on the target surface;The process is considered isothermal in the sense that the gas environment and all internal surfaces of the vacuum chamber have the same temperature;The model equations describe only the process on the target; there is no separate equation for the chamber wall and the balance equation for gas particles.

In [42], when modeling reactive sputtering, adsorption of the reactive gas on the wall of the vacuum chamber was physically correctly taken into account. Remaining within the framework of the isothermal model, the author expressed the formation of the compound on the target through adsorption, writing the balance equation for gas flows as follows:(4)$$Q_{0}=Q_{t}^{+}-Q_{t}^{-}+Q_{w}+Q_{p}.$$

In Equation (4), the reactive gas flows on the target surface due to adsorption and sputtering are denoted by *Q*_t_^+^ and *Q*_t_^−^, respectively; on the chamber wall—*Q*_w_; due to the operation of the vacuum pump—*Q*_p_. Next, the values of the effective pumping speed for each sorbing surface were introduced. The resulting equations allowed the author to construct kinetic curves and describe the hysteresis effect during the deposition of a TiO_2_ film. However, the kinetic dependences presented in [42] have inflection points; this does not correspond to the experimental results presented, for example, in [43]. Another feature of the result obtained in [42] is the possibility of a steady state with an intermediate value 0 < θ_t_ < 1. This contradicts all known experimental results.

In [44,45], in order to develop the model proposed in [21], the change in the partial pressure of the reactive gas was expressed in terms of gas flows:(5)$$\frac{dp}{dt}=-\frac{kT}{V}(\sum_{i}F_{i}A_{i}-Q_{0})-\frac{pS_{p}}{V},$$
where *V* is the vacuum chamber volume; *F_i_* is the density of the reactive gas flow on the *i-*th surface (target t, substrate s, chamber wall w); *A_i_* is the area of the *i*-th surface (*i* = t, s, w) consuming the reactive gas, the flow of which is equal to:(6)$$Q_{i}=\alpha F(1-\theta_{i})A_{i},$$
where θ*_i_* is the degree of coverage of the *i*-th surface by the compound. As a result, the balance equation for gas flows for the steady state appeared:(7)$$Q_{0}=\sum_{i}F_{i}A_{i}+\frac{pS_{p}}{kT}.$$
where the term $pS_{p}/kT$ defines the flow that passes through the pump. On the basis of expressions (5)–(7) and the low-pressure gas discharge model, the kinetics of Al sputtering in the Ar + O_2_ environment was studied in [44,45].

Thus, in [37,42,44,45], the reactive sputtering model was further developed. In its analytical description, equations for gas flows to all surfaces of the chamber and the balance equation for gas particles appeared.

Further development of reactive sputtering modeling was carried out in a series of papers [35,36,46,47,48,49,50,51,52]. The authors most consistently developed a model that we will call “general”, in contrast to the specific models of the previous section.

## 3. General Isothermal Chemisorption Model (the Berg Model)

### 3.1. Initial Model

The model that the authors described in [46] was later called the Berg model. It should be noted that the group of Professor S. Berg has been working hard on the development of the reactive sputtering model over the past thirty-five years and has published more than a dozen articles on this topic.

Further, when discussing different models, we use notations that partially differ from those adopted by the authors of the referred works and used in Section 1 of this article. This was done in order to unify the mathematical descriptions of different models.

In the first work of 1987 [46], the ideas expressed in the mentioned works were developed for the reactive sputtering of a one-component metal target M in the environment of one reactive gas X_2_. Describing the physical model, the authors proposed a number of constraints:The model should take into account the processes occurring on the surfaces of the target and the wall of the vacuum chamber with areas *A*_t_ and *A*_w_, respectively (the area of the substrate *A*_s_ was included in *A*_w_ due to its insignificance);The sputtering process occurs under isothermal conditions. This means that the target and the wall have the same temperature, *T*, which is equal to the temperature of the gas environment;Two processes compete on the target surface:Formation of the M*_m_*X*_n_* compound due to chemisorption with the kinetics (*d*θ_t_/*d*t)_chem_ (see Figure 2). Quantitatively, it is characterized by the sticking coefficient α, which specifies the fraction of the adsorbed gas flow with density *J*_X2_ incident on the surface (in Formulas (3) and (5)–(7), this value is denoted by *F*):
(8)$$J_{X_{2}}=\frac{p_{X_{2}}}{\sqrt{2\pi m_{X_{2}}kT}},$$
where *p*_X2_ and *m*_X2_ are the partial pressure and mass of a reactive gas molecule, respectively;Sputtering of the M*_m_*X*_n_* compound by argon ions with kinetics (*d*θ_t_/*d*t)_sp_.Chemisorption on surfaces covered by the M*_m_*X*_n_* compound is negligible;Sputtering of the M*_m_*X*_n_* compound from the target surface occurs in the form of molecules that are deposited on the walls of the chamber;The process has two independent variables: the discharge current density of argon ions on the target *j* and the *Q*_0_ reactive gas incoming flow (the flow introduced into the vacuum chamber).

The physical model presented above in items 1–6 was described by a series of algebraic equations. Two of them set the kinetics of processes on selected surfaces. For the target surface, the kinetic equation similar to Equation (1) is set:(9)$$\frac{d\theta_{t}}{dt}\;\;=\;\;{\left(\frac{d\theta_{t}}{dt}\right)}_{\mathrm{chem}}+\;\;{\left(\frac{d\theta_{t}}{dt}\right)}_{\mathrm{sp}},$$
where
(10)$${\left(\frac{d\theta_{t}}{dt}\right)}_{\mathrm{chem}}=2\alpha J_{X_{2}}\left(1-\theta_{t}\right)\;\;\;\mathrm{and}\;\;\;{\left(\frac{d\theta_{t}}{dt}\right)}_{\mathrm{sp}}=\left(\frac{j}{e}\right)S_{C}\theta_{t},$$

For the steady state, Equation (9), taking into account (10), has the form:(11)$$2\alpha J_{X_{2}}\left(1-\theta_{t}\right)-\left(\frac{j}{e}\right)S_{C}\theta_{t}=0.$$

During target sputtering, two fluxes arise (see Figure 2): M atoms with density *J*_tM_ and M*_m_*X*_n_* molecules with density *J*_tC_, which are deposited on the wall surface. Sputtering is absent there, and the kinetic equation for it was written as:(12)$$\frac{d\theta_{w}}{dt}\;\;=\;\;{\left(\frac{d\theta_{w}}{dt}\right)}_{\mathrm{chem}}+\;\;{\left(\frac{d\theta_{w}}{dt}\right)}_{\mathrm{spC}}-\;\;{\left(\frac{d\theta_{w}}{dt}\right)}_{\mathrm{spM}},$$
where the term (*d*θ_w_/*d*t)_chem_ describes the increase in the fraction of the wall surface θ_w_ covered by the compound film, resulting from chemisorption (see Figure 3); (*d*θ_w_/*d*t)_spC_ describes the increase in θ_w_ due to the flux of M*_m_*X*_n_* molecules sputtered from the target surface and falling onto a fraction of the wall covered by metal; (*d*θ_w_/*d*t)_spM_ describes the decrease in θ_w_ due to the flux of M atoms sputtered from the target surface and falling onto a fraction of the wall covered by the M*_m_*X*_n_* compound.

For the stationary state of the sputtering process, Equation (12), written taking into account all the details, takes the form
(13)$$2\alpha J_{X_{2}}\left(1-\theta_{w}\right)+\left(\frac{j}{e}\right)S_{C}\theta_{t}\left(\frac{A_{t}}{A_{w}}\right)\left(1-\theta_{w}\right)-\left(\frac{j}{e}\right)S_{M}\left(1-\theta_{t}\right)\left(\frac{A_{t}}{A_{w}}\right)\theta_{w}=0,$$

In addition to (11) and (13), three equations define the reactive gas flows in the sputtering system. Two of them are similar to (6) at *F* = *J*_X2_ with subscripts for the target surfaces *i* = t and for the wall *i* = w. In addition, a part of the gas *Q*_p_ is pumped out by a vacuum pump at a speed *S*_p_:(14)$$Q_{p}=pS_{p}.$$

The sixth equation of the analytical description of the reactive sputtering model, which is illustrated in Figure 4, sets the balance of gas flows:(15)$$Q_{0}=Q_{t}+Q_{w}+Q_{p},$$

The numerical solution of the system of Equation (6) for *i* = t and w, (11), (13)–(15) allowed the authors of [46] to determine the dependences *p* = *f*(*Q*_0_) at *j* = const. The authors qualitatively demonstrated the adequacy of the model for sputtering a titanium target in a nitrogen-containing environment.

In subsequent works by these and other authors [32,36,47,48,49,50,51,52,53,54,55,56,57,58,59,60,61,62], the initial isothermal chemisorption model was used to describe the processes of film deposition of various compounds. In addition, it was developed for more complex cases of sputtering of two targets in an Ar + X_2_ mixture and one target in an Ar mixture with two reactive gases.

### 3.2. Development of the Model

Developing the initial model, the authors in [47] proposed to take into account the reactive gas consumption on the full surfaces of the target and the chamber wall. This meant that gas adsorption was also allowed on parts of the surfaces covered by the compound. In the model with this assumption, the authors did not use the wall, replacing it with a substrate with area *A*_s_. In essence, under isothermal conditions, the name of the surface on which the particles of the substance sputtered from the target surface are deposited does not matter. It is important that its area has a significant effect on the sputtering process. Therefore, if in [46] the authors assumed *A*_w_ >> *A*_s_, then in [47], their obvious assumption was *A*_w_ << *A*_s_. In this regard, further, when writing the corresponding equations, the subscript symbol w will be replaced by the symbol s. Taking into account the introduced assumption, the equations for reactive gas flows on the surface of the target and substrate of type (6) take the form
(16)$$Q_{t}=[\alpha_{\mathrm{tM}}J_{X_{2}}(1-\theta_{t})+\alpha_{\mathrm{tC}}J_{X_{2}}\theta_{t}]A_{t},$$
(17)$$Q_{s}=[\alpha_{\mathrm{sM}}J_{X_{2}}(1-\theta_{s})+\alpha_{\mathrm{sC}}J_{X_{2}}\theta_{s}]A_{s},$$
where α_tM_, α_tC_, α_sM,_ and α_sC_—adsorption coefficients of reactive gas atoms to metal M and compound M*_m_*X*_n_* on the target and substrate surfaces, respectively.

Obviously, at α_tC_ = α_sC_ = 0.01, the reactive gas consumption will increase by no more than 2%. However, the equations describing the steady states of both surfaces of the type (11) and (13) cannot change since the values of θ_t_ and θ_s_ are influenced by the sorption of X_2_ gas only on the metal (fractions of the target surface 1 − θ_t_ and 1 − θ_s_). In other words, in the equations of the steady state of both surfaces given in [47], the terms written in the accepted notation
$$2\alpha_{\mathrm{tC}}J_{X_{2}}\left(1-\theta_{t}\right)\;\mathrm{and}\;2\alpha_{\mathrm{sC}}J_{X_{2}}\left(1-\theta_{s}\right),$$
can be discarded.

In [47,48], the authors made a very useful suggestion. They extended the analytical description of the model with an expression for the target sputtering rate:(18)$$R=\left(\frac{j}{e}\right)[S_{C}\theta_{t}+S_{M}(1-\theta_{t})].$$

The family of curves *R = f* (*Q*_0_)*_j_*_ = const_ is qualitatively shown in Figure 5. The authors assumed that the points corresponding to the deposition modes of a constant composition film lie in the *R* − *Q*_0_ plane on a straight line. Therefore, if we add straight lines coming from the origin of coordinates to the family *R* = *f* (*Q*_0_)*_j_*_ = const_, then each of them will represent the deposition process of the constant composition film. Figure 5 shows one of these straight lines corresponding to a film of stoichiometric composition (for θ_t_ = 1). A very important conclusion follows from this hypothesis: the growth rate of a film of a given composition can be changed by varying the discharge current, but it is also necessary to change the value of *Q*_0_ in such a way that the operating point remains on the corresponding straight line in the *R* − *Q*_0_ plane.

The work [49] was the next step in the development of the Berg model. In this work, the authors proposed models of two reactive sputtering processes. In the first one, the target of the two-element metal alloy *y*M_1_ + (1 − *y*)M_2_ was sputtered in the mixture of Ar + X_2_.

Figure 6 and Figure 7 illustrate the model of this process proposed by the authors. Processes of forming and sputtering M_1_X and M_2_X films compete on the target containing M_1_ and M_2_ metals (Figure 6). Fractions of the targe surface covered by the above films are θ_tC_1__ and θ_tC_2__, respectively. In this problem, the equation of the target steady state can be written in the form:(19)$$\alpha_{M_{1}}yJ_{X_{2}}\left(1-\theta_{{\mathrm{tC}}_{1}}\right)+\alpha_{M_{2}}\left(1-y\right)J_{X_{2}}\left(1-\theta_{{\mathrm{tC}}_{2}}\right)=\left(\frac{j}{e}\right)\left(yS_{C_{1}}\theta_{{\mathrm{tC}}_{1}}+(1-y)S_{C_{2}}\theta_{{\mathrm{tC}}_{2}}\right)$$
where *J*_X2_ is the reactive gas flow density, which is defined by an expression like (8); α_M_1__ and α_M_2__ are the adhesion coefficients of X_2_ molecules to fractions of the target surface where there are no M_1_X and M_2_X films, respectively; θ_tM_1__ and θ_tM_1__ are the fractions of the target surface where there are no M_1_X and M_2_X films, respectively; *S*_C_1__ and *S*_C_2__ are the sputtering yields of M_1_X and M_2_X films, respectively. The flux sputtered from the target surface consists of fluxes of atoms M_1_, M_2_ and molecules M_1_X, M_2_X with the corresponding densities, which are denoted by *J*_tM_1__, *J*_tM_2__, *J*_tC_1__, and *J*_tC_2__ in Figure 6. The same fluxes fall onto the substrate; their densities are denoted by *J*_sM_1__, *J*_sM_2__, *J*_sC_1__, and *J*_sC2_ in Figure 7. Other equations of the analytical description of the model are compiled by analogy with Equations (13)–(17).

In the second process, the co-sputtering of two targets made from different metals was studied. The physical model of this process is illustrated in Figure 8. The targets of metals M_1_, M_2_ are sputtered in the mixture of Ar + X_2_. As in all previous cases, the processes of formation and sputtering of compound films compete on their surfaces. The processes occurring on the targets are not related; therefore, in the steady state, two equations arise with two independent variables:(20)$$\alpha_{M_{1}}J_{X_{2}}\left(1-\theta_{{\mathrm{tC}}_{1}}\right)=\left(\frac{j}{e}\right)S_{C_{1}}\theta_{{\mathrm{tC}}_{1}}\;\mathrm{and}\;\alpha_{M_{2}}J_{X_{2}}\left(1-\theta_{{\mathrm{tC}}_{2}}\right)=\left(\frac{j}{e}\right)S_{C_{2}}\theta_{{\mathrm{tC}}_{2}}$$

The rules for compiling other equations of a system describing reactive sputtering of this type should not cause difficulties. The derivation of these equations is similar to the procedures used by Professor Berg and co-workers in the publications we reviewed earlier.

In this work, the authors admitted that they could not reliably demonstrate the adequacy of the proposed model for the case in which Ti, Al, and the Ar + O_2_ gas mixture were used. In addition, they admitted that neither theoretically nor experimentally it was possible to detect any individual influence of the used metals on the dependencies *p* = *f*(*Q*_0_). Let us note that the assumption of the authors that the composition of the metal alloy film deposited on the substrate will always be identical to the bulk composition of the target is not always valid. In some cases, the effect of preferential sputtering occurs, in which the target surface is continuously depleted of a component with a higher sputtering yield [63].

The solution to an interesting problem in the development of the chemisorption model is given in [52]. In contrast to [46], the authors complicated the process of reactive sputtering by placing one metal target in an Ar + O_2_ + N_2_ gas mixture. In the model of this process, as in [47], there is also no wall, and the following main assumptions were made:Two processes compete on the target surface:Formation of the MO + MN compound due to chemisorption. The process was quantitatively characterized by the sticking coefficients of gases to the metal α_O_ and α_N_, the molecules of which fall on the target surface with flux densities *J*_O_2__ and *J*_N_2__, each of which is given by Formula (8);Sputtering of the MO + MN compound with argon ions.Sputtering from the target surface of the MO + MN compound occurs in the form of MO and MN molecules, which are uniformly deposited on the substrate (the chamber wall is excluded from the model);The process has three independent variables: the discharge current density of the argon ions on the target *j* and incoming flows of reactive gases *Q*_O_2__ and *Q*_N2_.

To facilitate understanding of the analytical description of the model, we present it in the notation already used above. They differ from the author’s notations in the referred article. The steady state of the target surface in the new problem is now described by two equations. The steady state of the fraction of the target surface θ_tC_ covered by oxide MO gives the equation:(21)$$2\alpha_{1}J_{O_{2}}\left(1-\theta_{{tC}_{1}}-\theta_{{tC}_{2}}\right)+2\alpha_{12}J_{O_{2}}\theta_{{tC}_{2}}-\left(\frac{j}{e}\right)S_{C_{1}}\theta_{{tC}_{1}}=0,$$
where θ_tC_2__is the fraction of the target surface covered by nitride; α_12_ is the coefficient describing the probability of replacing the nitrogen with the oxygen in MN, as the metal–oxygen bond is energetically more favorable than the metal–nitrogen bond; *S*_C_1__ is the oxide sputtering yield. For a fraction of the target surface θ_tC_2__ covered by MN nitride, the following holds true in the steady state:(22)$$2\alpha_{2}J_{N_{2}}\left(1-\theta_{{tC}_{1}}-\theta_{{tC}_{2}}\right)-2\alpha_{12}J_{O_{2}}\theta_{{tC}_{2}}-\left(\frac{j}{e}\right)S_{C_{2}}\theta_{{tC}_{2}}=0,$$
where *S*_C_2__ is the sputtering yield of nitride.

The steady state of the substrate surface in this problem is described by two equations similar to (13), where the subscript w is replaced by s, separately for fractions covered by oxide θ_sC_1__ and nitride θ_sC2_.

The flows of reactive gases in the sputtering system set:Oxygen flows on the surfaces of the target *i* = t and wall *i* = w:
(23)$$Q_{iO_{2}}=\alpha_{1}J_{O_{2}}(1-\theta_{iC_{1}}-\theta_{iC_{2}})A_{i}+\alpha_{12}J_{O_{2}}\theta_{iC_{2}}A_{i};$$Nitrogen flows on the surfaces of the target i = t and wall i = w:
(24)$$Q_{iN_{2}}=\alpha_{N}J_{N_{2}}(1-\theta_{iC_{1}}-\theta_{iC_{2}})A_{i}-\alpha_{12}J_{O_{2}}\theta_{iC_{2}}A_{i},c^{-1};$$Gas flows, which are pumped out:
(25)$$Q_{\mathrm{pO}}{}_{2}=p_{O_{2}}S_{p},\;\mathrm{and}\;Q_{{\mathrm{pN}}_{2}}=p_{N_{2}}S_{p},$$
where *S*_p_ is a pumping speed of the vacuum chamber (does not depend on the type of gas);Balance equation for flows:
(26)$$Q_{0O_{2}}=Q_{{\mathrm{tO}}_{2}}+Q_{{\mathrm{sO}}_{2}}+Q_{{\mathrm{pO}}_{2}}\;\mathrm{and}\;Q_{0N_{2}}=Q_{{\mathrm{tN}}_{2}}+Q_{{\mathrm{sN}}_{2}}+Q_{{\mathrm{pN}}_{2}}.$$

Thus, the analytical description of the physical model of the reactive sputtering of a single metal target in an Ar + O_2_ + N_2_ mixture contains 10 equations. Calculations performed by the authors using expressions (19)–(26) made it possible to establish that the addition of nitrogen to the gas environment contributes to a decrease in the hysteresis effect up to its complete disappearance. In addition, the rate of film deposition increases in comparison with the sputtering of oxides without the addition of nitrogen.

An experimental confirmation of the effects predicted by the model in [52] was performed in [53]. The authors showed that the calculated dependences of the partial pressures of both reactive gases on the incoming nitrogen flow are close to those found experimentally.

In [60], an even more complicated problem of reactive sputtering of two targets (aluminum and zirconium) in an Ar + O_2_ + N_2_ mixture was studied. Detailed coverage of the evolution of the initial model allows for refraining from presenting the details of [60] here. It is obvious that a second target has appeared in the physical model of this process. Two flows of reactive gases fall on each target in the same way as for a single target in the previous problem. Therefore, in the analytical description of the model, equations of the steady state of each target arose. Fluxes of particles sputtered from their surfaces are deposited on the substrate, which leads to a corresponding change in the equation of its steady state. According to the authors, the model is in qualitative agreement with the experimental results.

In [64], an improved model for reactive sputtering in a vacuum chamber with an Ar + X_2_ mixture is discussed. The magnetron, equipped with an aluminum target, is fixed at the top. The Berg simple isothermal chemisorption model was refined by expanding the list of assumptions:The target is sputtered only in an annular area *A*_r_, called the racetrack. Its shape and position on the target are determined by the magnetic field;Particles sputtered from the target surface are deposited on:A substrate with area *A*_s_ located in the center part of the bottom of the chamber;The peripheral part of the bottom with area *A*_pb_;The chamber walls with area *A*_w_;The top of the chamber with area *A*_top_, including the non-sputtered part of the target.There is no redeposition of sputtered particles in the upper part of the chamber.

To describe the steady state of the target in this problem, Equation (11) can be used, replacing the variable θ_t_ with θ_r_. For a similar state of the *i*-th (*i* = s, pb, w, and top) surface on which the sputtered particles are deposited, the following equation should be used:(27)$$2\frac{m}{n}\alpha J_{X_{2}}\left(1-\theta_{i}\right)A_{i}+m\left(\frac{j}{e}\right)S_{C}\theta_{r}f_{i}\left(1-\theta_{i}\right)A_{r}=\left(\frac{j}{e}\right)S_{M}\left(1-\theta_{r}\right)f_{i}\theta_{i}A_{r},$$
which is constructed by analogy with Equation (13). The main part of the notation in (27) was used by us earlier. In addition, in (27), it is assumed: θ*_i_* is the fraction of the area of the *i*-th (*i* = s, pb, w, and top) surface covered by the M*_m_*X*_n_* compound; θ_r_ is the fraction of the area of the sputtered annular target area covered by the M*_m_*X*_n_* compound; *f_i_* is the fraction of the material sputtered from the target that is deposited on the *i*-th (*i* = s, pb, w, and top) surface.

Reactive gas flows to selected surfaces are given by equations of the form (6):(28)$$Q_{i}=\alpha J_{X_{2}}(1-\theta_{i})A_{t}$$
when the gas balance is equal to
(29)$$Q_{0}=\sum_{i}Q_{i}+pS_{p}$$

In all previous modeling works, the authors used the incoming flow of the reactive gas and the discharge current as independent variables. In [64], it was proposed to supplement this list with the distance *d* between the target and the substrate. Applying the new parameter changed the steady state equation for each *i*-th (*i* = s, pb, w, and top) surface:(30)$$\begin{array}{l} 2\frac{m}{n}\alpha J_{X_{2}}\left(1-\theta_{i}\right)A_{i}+m\left(\frac{j}{e}\right)S_{C}\theta_{r}f_{0}{\left(\frac{d_{0}}{d}\right)}^{2}\left(1-\theta_{i}\right)A_{r}\\ \;\;\;\;\;\;\;\;\;\;\;\;=\left(\frac{j}{e}\right)S_{M}\left(1-\theta_{r}\right)f_{0}{\left(\frac{d_{0}}{d}\right)}^{2}\theta_{i}A_{r}\; \end{array},$$
where *f*_0_—ratio of the sputtered flux in the target plane to the scattered flux at a distance *d*_0_ from the target.

In conclusion of the discussion of work [64], in which Equations (27)–(30) are the main ones, we note that it presents the next step in the development of the reactive sputtering model. For the first time, the surface of the substrate appeared in it as an independent object. If only this innovation is accepted, excluding other surfaces, then two more equations should be added to the system of equations of the initial model containing six equations. The first of them should describe the steady state of the substrate surface. It can be obtained from Equation (13) by replacing the subscript w with the index s. The second equation for the gas flow consumed by the substrate can be obtained from (6) at *i* = s.

### 3.3. Adequacy of the Initial Model

Until now, the Berg initial isothermal chemisorption model has been a popular tool for studying the processes of film deposition of simple compounds with the reactive sputtering method [65,66,67]. Undoubtedly, it was an advance in the modeling of the reactive sputtering process. Its analytical description allows us to evaluate the relationship between the process parameters and shows that a hysteresis can exist.

However, as noted in [65], this model has very rough assumptions. In agreement with this statement, let us pay attention to only two circumstances. First, the deposited film, in terms of its physical properties, is a chemical compound that is formed on the substrate mainly due to the plasma–chemical reaction occurring on the target. The replacement of the reaction by the mechanism of chemisorption should be recognized as a very strong simplification. The second strong simplification is the assumption that the temperatures of all surfaces are equal. In real processes, the surfaces of the target, substrate, and chamber walls have different temperatures: the target surface can be heated up to 700–900 K, the substrate is heated above 600 K, and the wall temperature can be maintained at 300 K. In addition, the substrate temperature can steadily increase during deposition due to the release of condensation energy, the kinetic energy of particles, discharge radiation, and also the energy released by chemical reactions [16,68,69,70].

Let us show by examples the possibilities of the initial Berg model. For this, we use the experimental results presented in [25,32]. In [32], the authors studied the deposition of Ta_2_O_5_ films at sputtering system parameters equal to *S*_p_ = 8.6 L/s; *A*_t_ = 100 cm^2^; *A*_w_ = 300 cm^2^; *j* = 0.05 A/cm^2^. The result of solving the system of Equation (6) at *i* = t and w, (12), (14)–(16) for this experiment at α = 1, *S*_C_ = 0.024, *S*_M_ = 0.6 is shown by the solid line in Figure 9a.

In [25], the authors studied the deposition of TiO_2_ films at the sputtering system parameters of *S* = 190 L/s; *A*_t_ = 190 cm^2^; *A*_w_ = 9000 cm^2^; *j* = 0.011 A/cm^2^. The result of solving the system of Equation (6) at *i* = t and w, (12), (14)–(16) for this experiment at α = 1, *S*_c_ = 0.016, *S*_m_ = 0.32 is shown by the solid line in Figure 9b. It can be seen from both figures that the Berg model does not adequately predict the points at which the operating modes of the target change.

The obtained results may indicate both the inadequacy of the initial Berg model and the inaccuracy of the values of the model parameters that were used in the calculation. Let us carry out a more detailed analysis of the example of an experiment on the deposition of the Ta_2_O_5_ film in [32].

Figure 9a shows that the analytical dependence gives overestimated values of the flow *Q*_0_ at both points of instability A and C. In addition, the linear section of the curve in the region of the reactive mode is significantly shifted relative to the experimental points and has a slight difference in the slope angle. The first difference indicates an overestimated gas flow to the target surface, which maintains the steady oxide mode of its operation. It significantly depends on the target area *A*_t_. The angle of inclination of this section reflects the operation of the vacuum pump only. The tangent of this angle is equal to 1/*S*_p_.

These differences between the experiment and the simulation result were eliminated by selecting adequate values of *A*_t_ and *S*_p_. Figure 10 shows the dependencies built on the basis of the changed data, the values of which are indicated in the text below the figure. Figure 10 demonstrates that the target area indicated in [32] is overestimated by about 2.9 times, and the pumping speed of the vacuum chamber is overestimated by 1.08 times. Here, it can be assumed that the authors of [32] took into account the total target area, which was used in calculating the current density under the assumption of its uniform distribution along the target surface. While the effective area involved in the process of reactive sputtering, which sets the configuration of the magnetic field, is significantly smaller.

We also note that the results shown for different experiments in Figure 9b and Figure 10 are qualitatively identical. They demonstrate that the points of instability, where the target operating modes change, are shifted to the region of lower values of the incoming gas flows. This may indicate the inadequacy of the Berg original model. Moreover, here, it is necessary to pay attention to the assumptions made in the formation of the physical model at the beginning of this section.

We retain all the approximations adopted in the model under consideration, except for the second one, in which the temperature of the target and the walls of the vacuum chamber are assumed to be equal. We denote them using *T*_t_ and *T*, respectively, assuming that the temperatures of the walls and the gas are the same. The substrate in this model is not separated into an independent object; its area is included in the wall area. Therefore, the substrate temperature is also considered to be equal to *T*, although in real processes, this parameter is set at a level of 600 K and higher, which significantly affects the film deposition process.

The separation of temperatures *T*_t_ and *T* is, in fact, a partial transition to a non-isothermal model. Characterizing chemisorption on different surfaces, we introduce the sticking coefficients for the target α_t_ and for the wall α_w_, assuming α_t_ ≠ α_w_. Let us assume that as the temperature increases, the sticking coefficient of reactive gases decreases. Therefore, when choosing the values of α_t_ and α_w_, two conditions must be met: α_t_ < 1, α_w_ < 1 and α_t_ < α_w_, as *T*_t_ > *T*.

Let us rewrite Equation (6) at *i* = t, w, (11) and (13), taking into account accepted assumptions and notations:(31)$$Q_{i}=\alpha_{i}J_{X_{2}}(1-\theta_{i})A_{i}.$$
(32)$$2\alpha_{t}J_{X_{2}}\left(1-\theta_{t}\right)-\frac{n}{m}\left(\frac{j}{e}\right)S_{C}\theta_{t}=0,$$
(33)$$2\alpha_{w}J_{X_{2}}\left(1-\theta_{w}\right)+\frac{n}{m}\left(\frac{j}{e}\right)S_{C}\theta_{t}\left(\frac{A_{t}}{A_{w}}\right)\left(1-\theta_{w}\right)=\left(\frac{j}{e}\right)S_{M}\left(1-\theta_{t}\right)\left(\frac{A_{t}}{A_{w}}\right)\theta_{w},$$

Based on the solutions of these equations, we can assume that the elimination of the isothermal condition from the initial Berg model makes it possible to increase its adequacy. However, another improvement of the reactive sputtering model is also acceptable. As Figure 10 shows, it is important to change the physical model in order to shift the transition point of the target from the metallic to the reactive mode to the right. To do this, it is necessary to increase the growth rate of the compound film by adding an additional mechanism of the film formation to the model.

### 3.4. New Mechanism

For the first time Professor Berg and co-workers in [71,72] with reference to [73], proposed to introduce an assumption about a significant influence of the implantation of reactive gas ions into the near-surface layer of the target on reactive sputtering. In [74], equations were given that describe a new model of reactive sputtering of a metal target in an Ar + X_2_ mixture. In [75], the updated Berg model is described in detail. The authors supplemented the initial model with new assumptions:The M*_m_*X*_n_* compound is sputtered from the target in the form of atoms;The formation of the M*_m_*X*_n_* compound on the target occurs not only on the surface due to the chemisorption of X_2_ molecules but also in the volume under the surface due to the interaction of free metal atoms with X atoms that penetrated into it;X atoms penetrated into the target volume due to the implantation of accelerated $X_{2}^{+}$ ions and adsorbed X atoms, which received momentum due to the impact with the Ar^+^ ion. The latter mechanism was called the “knock-on effect”;Sputtering of the target by $X_{2}^{+}$ ions is negligible.

The target in the updated model only serves as a source of the metal flux with density *J*_tM_ (Figure 11). At the same time, the kinetic Equation (9) acquired new components:(34)$$\frac{d\theta_{t}}{dt}={\left(\frac{d\theta_{t}}{dt}\right)}_{\mathrm{chem}}+{\left(\frac{d\theta_{t}}{dt}\right)}_{\mathrm{impl}}+{\left(\frac{d\theta_{t}}{dt}\right)}_{\mathrm{sput}}+{\left(\frac{d\theta_{t}}{dt}\right)}_{\mathrm{kn}},$$
where (*d*θ_t_/*dt*)_impl_ is the growth rate of the M*_m_*X*_n_* compound film due to ion implantation; (*d*θ_t_/*dt*)_kn_ is the rate of removal of reactive gas atoms due to the knock-on effect.

For the steady state, Equation (34) takes the form
(35)$$2\alpha J_{X_{2}}\left(1-\theta_{t}\right)+2\alpha_{i}\left(\frac{j}{e}\right)\left(\frac{p}{p_{A}+p}\right)\left(1-\theta_{t}\right)-\left(\frac{j}{e}\right)S_{C}\theta_{t}-\left(\frac{j}{e}\right)S_{k}\theta_{t}^{2}=0,$$
where *p* is the partial pressure of the reactive gas; *p*_A_ is the partial pressure of Ar; *S*_k_ is a coefficient similar to the sputtering yield, which determines the number of X atoms implanted into the target volume due to impact with one Ar^+^ ion; α_i_ is the probability of implantation of a reactive gas ion into the target volume. The rest of the designations in (34) were used earlier. The arrangement of terms in (35) corresponds to formula (34). This makes it possible to understand the physical meaning of each of them. For example, the first term describes the formation of a film on the target surface due to chemisorption, the second term—due to implantation, and so on.

In this problem, only fluxes of reactive gas molecules and metal atoms with densities specified below fall on the substrate. In this case, the respective densities are *J*_X2_ (see (8)) and
(36)$$J_{\mathrm{tM}}=\left(\frac{j}{e}\right)\left[S_{\mathrm{MC}}\theta_{t}+S_{M}\left(1-\theta_{t}\right)\right],$$
where *S*_MC_ is a coefficient determining the partial sputtering of metal atoms from the M*_m_*X*_n_* layer. Taking into account (36), the stationary equation for the substrate becomes simpler than (13):(37)$$2\alpha J_{X_{2}}\left(1-\theta_{s}\right)-J_{\mathrm{tM}}\left(\frac{A_{t}}{A_{s}}\right)\theta_{s}=0,$$

As in all the models considered here, the system of equations describing the physical model should include equations for reactive gas flows to the target and substrate, which can be used as (31), the pumping Equation (14), and the gas balance equation of the type (15).

In conclusion, the authors noted that the inclusion of atomic M*_m_*X*_n_* sputtering and implantation in the initial Berg model was intended to expand the number of modeling parameters. However, from a practical point of view, these interesting proposals complicate the modeling since, in each case, they require a search for the values of these parameters.

In addition, in [75], the authors described a two-layer model based on a more complex dynamical model, which they described earlier in [74]. In this publication, the authors develop the Berg model by assuming that the compound layer on the target surface is several tens of angstroms thick. In the new model, this continuous layer is represented as *N* discrete layers.

In [76], the authors applied a new model to study the hysteresis. Studies of the influence on the dependence *p* = *f*(*Q*_0_) of various model parameters allowed them to establish the possibilities for reducing or completely eliminating hysteresis.

Let us pay attention to one more work by Berg et al. [54]. When sputtering a titanium target in an Ar + CH_4_ mixture, the authors found that depending on *Q*_0_, a film with a carbon content from 0 to 100% can be formed on the substrate. The authors expected similar behavior when carrying out the processes in Ar + B_2_H_6_ and Ar + Si*_n_*H_2*n*+2_. For such processes, which the authors called “non-saturated”, the initial Berg model turned out to be unsuitable.

The new model proposed by the authors will be explained with the help of Figure 12 and Figure 13. On the target (Figure 12), in contrast to the initial model, the processes of formation and sputtering of TiC and C films compete. In accordance with this, the steady state equation of the target takes the form
(38)$$\alpha_{\mathrm{Ti}}J_{{\mathrm{CH}}_{4}}\left(1-\theta_{\mathrm{tTiC}}-\theta_{\mathrm{tC}}\right)+\alpha_{\mathrm{TiC}}J_{{\mathrm{CH}}_{4}}\theta_{\mathrm{tTiC}}+\alpha_{C}J_{{\mathrm{CH}}_{4}}\theta_{\mathrm{tC}}=\left(\frac{j}{e}\right)\left(S_{\mathrm{TiC}}\theta_{\mathrm{tTiC}}+S_{C}\theta_{\mathrm{tC}}\right),$$
where *J*_CH4_ is the methane flux density, which is defined by an expression like (8); α_Ti_, α_TiC_, and α_C_ are the adhesion coefficients of methane molecules to parts of the target surface with open titanium and covered by titanium carbide TiC and carbon C, respectively; θ_Ti_, θ_TiC_, and θ_C_ are the fractions of the target surface with open titanium and covered by carbide TiC and carbon C, respectively; *S*_TiC_ and *S*_C_ are the sputtering yields of titanium and carbon carbide films, respectively. The flux sputtered from the target surface consists of fluxes of Ti and C atoms and TiC molecules with the corresponding densities, which are denoted by *J*_tTi_, *J*_tC_, and *J*_tTiC_ in Figure 12. The same fluxes fall onto the substrate with densities, which are denoted by *J*_sTi_, *J*_sC_, and *J*_sTiC_ in Figure 13. The equation of the steady state of the substrate in this problem, as well as other equations of the system, are compiled by analogy with Equations (13)–(17). The model developed in this way, according to the authors, is adequate for the experimental results.

In a recent paper [77], Berg et al. proposed a model of reactive high-power pulsed sputtering, demonstrating it using the example of Ti sputtering in an Ar + O_2_ mixture. The system of equations describing the proposed model is similar to the system for the initial model and contains equations of the type (6), (11), and (13)–(15).

Ease of learning and use provided the Berg model with multilateral interest in the last twenty years. The isothermal chemisorption model was used by experts to describe practical problems [78,79,80,81,82,83,84,85]. Some of them suggested options for its development [86,87,88,89,90]. Among publications on modeling of reactive sputtering, there were works describing other models based, for example, on the laws of thermodynamics [91,92], statistics [93], etc. [94,95]. A tutorial [65] and review articles [96] on this subject have been published.

## 4. Isothermal Model of Depla (the RSD Model)

Let us draw the reader’s attention to the work of Professor Depla and his group. This team has been actively involved in studies of reactive sputtering and its modeling for the last twenty years. In terms of the intensity of publications in this area, the group from Ghent University can confidently be given second place after the group of Professor Berg. The results of experimental studies of reactive sputtering and the properties of the obtained films of oxides, nitrides, etc., were published in articles [73,97,98,99,100,101,102,103,104,105,106,107,108,109,110,111,112,113,114,115]. Various aspects of reactive sputtering modeling are described by the researchers of the team in publications [65,116,117,118,119,120,121,122,123,124,125,126,127,128,129,130,131,132,133,134,135,136,137,138,139]. Let us take a look at the main ones.

### 4.1. Initial RSD Model

The work [116] provides data on the influence of the incoming oxygen flow on the change in voltage on an aluminum target sputtered in an Ar + O_2_ mixture. In this article, the authors, for the first time, put forward the hypothesis that chemisorption alone cannot explain the modification of the target surface. The observed changes in the discharge voltage, in their opinion, can be additionally initiated by the chemical reaction of target oxidation in the near-surface layers by implanted oxygen atoms. Implantation and subsurface bulk response can be seen as the most important extension of the Berg original model. Its inclusion in the reactive sputtering model led to the appearance of the original model, which in [127] was called the RSD (Reactive Sputtering Deposition) model.

The same hypothesis was put forward in [73], devoted to the study of reactive sputtering of a copper target in Ar + N_2_. A similar conclusion was made in [97] based on the results of a study of silicon target sputtering in the same mixture. Detailing the process, the authors in [97] suggested that molecular nitrogen ions $N_{2}^{+}$ are implanted into the target, which, losing their charge, dissociate into individual atoms that react with the target material to form the M*_m_*X*_n_* compound. Based on these studies, the physical model of reactive sputtering was supplemented by implantation.

In order to form the rationale for their hypothesis about implantation, the authors referred to the results of a study from [140]. In this article, using the method of nuclear reactions, the reactive sputtering of a titanium target in a nitrogen-containing environment was studied.

It was found that the maximum number of retained nitrogen atoms after turning off the magnetron significantly exceeds the adsorbed monolayer. The result obtained, according to the authors, is associated with the implantation of nitrogen molecular ions with subsequent decay into atoms. In addition, the authors assumed that there was an implantation of atoms from adsorbed nitrogen molecules, which received a direct impact from the accelerated argon ion. Such particles that have arisen during ion bombardment of a target are called recoil atoms. Having received an impulse, they can leave the target or create a cascade of collisions in its surface layers. Subsequently, the mechanism of implantation of recoil atoms was called the “knock-on effect” in the model of Depla. The results of the experiment were confirmed by computer simulation.

In [117,118], the authors took the first steps in developing a new model. In [117], the influence of only implanted atoms, which enter into chemical interaction with target atoms, on the process was shown.

In essence, the RSD model can be expressed by the kinetic Equation (34), although it is not written explicitly in Depla’s works. The first attempt to write such an equation was made in [117] in the form
(39)$$n_{0}\frac{d\theta_{\mathrm{tC}}}{dt}=\frac{1}{n}\alpha_{\mathrm{tC}}\left(1-\theta_{\mathrm{tC}}\right)\frac{dn\left(0,\;t\right)}{dt},$$
where *n*_0_ is the target material density; *n* is the number of gas atoms in the compound; α_tC_ is the probability of the chemical reaction between implanted atoms X and target M; θ_tC_ the degree of surface coverage by the M*_m_*X*_n_* compound; *n*(0, *t*) is the surface concentration of implanted atoms. Equation (39) describes a model of the sputtering process based only on the implantation of ions $X_{2}^{+}$ into the target. In (39) there is no need to take into account sputtering of the compound by argon ions. Indeed, *d*θ_tC_/*dt* is the effective compound formation rate since the sputtering effect was taken into account in the calculation of *dn*/*dt*. In (39), the chemical reaction is given in the simplest form, which does not take into account the law of mass action and the Arrhenius equation. It is set using the coefficient α_tC_. In subsequent works, the authors complicated this part of the model.

In [118], the chemisorption mechanism from the Berg initial model was added to the model that takes into account only ion implantation. Thus, volumetric and surface mechanisms of M*_m_*X*_n_* formation appeared in the model, which can be expressed by the equation:(40)$$\frac{d\theta_{t}}{dt}={\left(\frac{d\theta_{t}}{dt}\right)}_{\mathrm{chem}}+{\left(\frac{d\theta_{t}}{dt}\right)}_{\mathrm{impl}}+{\left(\frac{d\theta_{t}}{dt}\right)}_{\mathrm{sput}}.$$

If in Equation (40), as in (39), we assume that sputtering is already taken into account in the second term on the right side, then the third term should be considered as sputtering of a surface film formed due to chemisorption.

This new model with the addition of the knock-on effect is described in more detail in [119]. It takes into account the processes occurring on the surfaces of the target and substrate. One of the system equations describing sputtering establishes, as in all previous cases, the balance of gas flows of the type (15). Three more equations define the gas flows to both surfaces of the type (6) and the pumped-out flow (14). The equations of the steady state of the target and the substrate are the root equations.

Let us consider the work [119] in more detail since it is the key publication of Professor Depla and his team on modeling. The authors detailed the main assumptions of their model in this article. Subsequently, some variations of the RSD model were described in numerous articles. These include the monograph [122] and the tutorial [65], which is a review. In these works, one can get acquainted in detail with both the process of reactive magnetron sputtering and its modeling.

Initially, we highlight the main assumptions of the model in a way that can be understood from studying Depla’s team publications. Using the assumptions described earlier in this work, the authors formed a physical model of the process, which was further described analytically using known physical or chemical laws, equations, and formulas. We believe that this is very important in order to see the essence of a particular model, as most often, its understanding is hindered for the reader by the dry lines of formulas.

Therefore, from the publication [119] and other articles of these authors, the RSD physical model can be expressed using a number of assumptions:(1)The process is isothermal, i.e., temperatures of all surfaces and gas are equal;(2)The process is accompanied by:(2.1)Direct implantation of X atoms into a target with a dose *D* ≈ 2*f j* + *t*/*e =* 2*j* + *tp*/*e*(*p* + *p*_A_), determined by the ion current density *j* and the molar fraction of the reactive gas in the vacuum chamber *f* (*e* is the electron charge);(2.2)Implantation of X atoms due to the destruction of M*_m_*X*_n_* molecules on the target surface by a direct impact of an Ar^+^ ion (knock-on effect with coefficient β representing the number of implanted atoms per ion in a flow with density *j*/*e*);(2.3)Chemisorption of X atoms resulting from the dissociation of adsorbed X_2_ molecules on the target surface determined by the sticking coefficient α.(3)The normalized distribution of projected ranges *c*(*x*) of ions $X_{2}^{+}$ along the normal to the target surface is a Gaussian:(41)$$c(x)=\frac{D}{\sqrt{2\pi }\Delta R_{p}}\exp\left[-\frac{{\left(x-R_{p}\right)}^{2}}{2\Delta R_{p}{}^{2}}\right],$$
where *R_p_* is the average ion range; Δ*R_p_* is the straggle;(4)The distribution of implanted atoms *n*(*x*, *t*) taking into account sputtering at a speed *v*_0_, is determined by the integral:(42)$$n(x,\;t)=2\int_{0}^{t}\frac{j}{e}fc\left(x-v_{0}\tau \right)d\tau .$$
where *c*(*x* − *v*_0_τ) is the normalized distribution taking into account sputtering;(5)The M*_m_*X*_n_* compound is formed:(5.1)On the surface due to chemisorption;(5.2)In the near-surface layers of the target due to the bulk chemical reaction *m*M + *n*X ↔ M*_m_*X*_n_*
between a part of implanted X atoms with metal atoms. The reaction rate is determined by the expression
(43)$$r=-\frac{n}{m}kn_{X}\left(x,\;t\right)n_{M}\left(x,\;t\right),$$
where *k* is the rate constant of the bulk reaction; *n*_X_ (*x*, *t*) is the concentration of unreacted implanted X atoms, taking into account sputtering (see expression (42)); *n*_M_(*x*, *t*) is the concentration of free target atoms.(6)The formation of the M*_m_*X*_n_* layer on the surface is accompanied by the process:(6.1)Sputtering of M*_m_*X*_n_* molecules with argon ions;(6.2)Removal of X atoms from M*_m_*X*_n_* molecules due to the knock-on effect.

Let us clarify that the M*_m_*X*_n_* compound, which occupies a fraction of the target surface θ_tC_, was formed in the subsurface layer due to a bulk chemical reaction with the participation of implanted X atoms. It appeared on the surface as a result of sputtering of the surface and subsequent layers. In fact, due to this process, the surface layer moves inside the target, constantly increasing the value of θ_tC_.

(7)In the area of the target surface, there are (Figure 14):(7.1)The layer from which the sputtering takes place directly (Surface). It has a thickness on the order of a monomolecular layer *s*. In the general case, this layer consists of sections: θ_tM_, which is free of the M*_m_*X*_n_* compound film; θ_tC_ and θ_tCch_, which are occupied by the M*_m_*X*_n_* compound formed due to direct implantation of X atoms and their chemisorption, respectively;(7.2)A subsurface region with a thickness *L* ≈ *R_p_* ± 3Δ*R_p_*, into which reactive gas atoms are implanted, chemically interacting with the target material. In the model, this area is represented as a thin layer called Subsurface, as shown in Figure 14. It consists of the θ_tCs_ region occupied by the M*_m_*X*_n_* compound formed due to the bulk chemical reaction, and 1 − θ_tCs_ region occupied by free atoms of metal M.

It was noted in [119] that chemisorption and implantation are identical in their effect on the formation of a compound film on a target. The second process increases the thickness of this film, which appears not only on the surface but also in near-surface layers with a thickness of a few nanometers. The rate of film growth on the surface also increases.

A new reactive sputtering model was presented in [119] in stages. In the first step, the authors took into account only the process of implantation of reactive gas ions, which is accompanied by a chemical reaction and sputtering. Under these conditions, θ_tCch_ = 0 (see Figure 14), and from the assumptions of the model, items 2.2, 2.3, 5.1, and 6.2. In this problem, as in all previous models, the main one is the kinetic equation describing the change in the state of the target surface. It is easy to write down simplifying (40):(44)$$\frac{d\theta_{t}}{dt}={\left(\frac{d\theta_{t}}{dt}\right)}_{\mathrm{impl}}.$$

However, in [119], the target was described taking into account expressions (40), (42), and (43) by the kinetics of bulk processes:(45)$$\begin{array}{l} \frac{\partial n_{X}\left(x,\;t\right)}{\partial t}=2\frac{j}{e}fc\left(x-v_{0}\;t\right)-\frac{n}{m}kn_{X}\left(x,\;t\right)n_{M}\left(x,\;t\right);\\ \;\;\;\;\;\;\;\;\frac{\partial n_{M}\left(x,\;t\right)}{\partial t}=-\frac{n}{m}kn_{X}\left(x,\;t\right)n_{M}\left(x,\;t\right). \end{array}$$

In this case the integral sputtering yield of the target was defined as:(46)$$\theta_{\mathrm{tC}}S_{C}+\theta_{\mathrm{tM}}S_{M}=S_{s}.$$
where θ_tC_ and θ_tM_ are the surface fractions covered by film and without it, respectively (see Figure 14); *S*_C_ and *S*_M_ are the partial sputtering yields of the M*_m_*X*_n_* compound molecules and M target atoms, respectively. It is obvious that θ_tC_ + θ_tM_ = 1.

The correctness of the first part of the model was verified by applying it to the results of studying the angular dependence of the oxidation degree of surface θ_tC_ and subsurface θ_tC_ layers of a silicon target under bombardment with only oxygen ions with an energy of 5 kev.

By solving the system of Equation (45), it was shown that the model based on ion implantation is able to adequately describe experiments of this type. This result was achieved by adjusting the reaction rate constant *k* = 1 × 10^−22^ cm^3^×c^−1^.

Further, in [119], using the model, the dependences of θ_tC_ and θ_tCs_ on *f* = *p*/(*p* + *p*_A_) were estimated for sputtering of a silicon target in an Ar + O_2_ mixture. It was found that as *f* increases, the dependences θ_tC_ = *f* (*f*) and θ_tCs_ = *f* (*f*) increase nonlinearly with saturation up to θ_tC_ = 1 and θ_tCs_ = 1. Similar results were also obtained for reactive sputtering of a silicon target in Ar + N_2_.

In the next step in [119], the RSD model was completely formed by including chemisorption and knock-on effect. In accordance with assumption 6 of the model in the steady state, three processes are involved in the formation of the target surface fraction θ_tCch_ covered by the M*_m_*X*_n_* compound:Formation of M*_m_*X*_n_* molecules on the fraction of the target surface θ_tM_ by chemisorption;Removal of M*_m_*X*_n_* molecules by sputtering;Removal of X atoms from M*_m_*X*_n_* molecules due to the knock-on effect. In fact, this process is also understood by the authors of the model as the removal of M*_m_*X*_n_* molecules.

The addition of the model with chemisorption and the knock-on effect required changing the first equation in system (45).
(47)$$\frac{\partial n_{X}\left(x,\;t\right)}{\partial t}=2\frac{j}{e}fc\left(x-v_{0}\;t\right)-\frac{n}{m}kn_{X}\left(x,\;t\right)n_{M}\left(x,\;t\right)+\frac{j}{e}\beta \theta_{{\mathrm{tC}}_{\mathrm{ch}}}p_{c}\left(x,\;t\right),$$
where *p*_c_(*x*) is the distribution of atoms implanted due to the knock-on effect. In the first approximation, it is assumed that this distribution is also a Gaussian.

After formulation of the equations describing the target state in the new model, the authors studied the influence of chemisorption and the knock-on effect on the degree of target oxidation during silicon sputtering in Ar + O_2_. It appeared that without the knock-on effect (β = 0), as the adhesion coefficient increases, the S-shaped dependence θ_rb_ on the reactive gas mole fraction *f* turns into an exponential with saturation and an inflection point. On the other hand, at a constant value of α = 0.01, an increase in β from 0 to 1 has the opposite influence on the degree of oxidation.

A detailed analysis of the target state in the new model was supplemented by the equation of the steady state of the substrate (wall), which differed from the analogous equation in the Berg chemisorption model (33) in one detail:(48)$$2\alpha_{i}J_{X_{2}}\left(1-\theta_{i}\right)+\frac{n}{m}\left(\frac{j}{e}\right)S_{C}\left[\theta_{{\mathrm{tC}}_{\mathrm{ch}}}+\theta_{\mathrm{tC}}\right]\left(\frac{A_{t}}{A_{i}}\right)\left(1-\theta_{i}\right)=\left(\frac{j}{e}\right)S_{M}\theta_{\mathrm{tM}}\left(\frac{A_{t}}{A_{i}}\right)\theta_{i},\;i=s,w.$$

This detail is visible in the second term on the left side of Equation (48). In square brackets, the fraction θ_tC_ with the M*_m_*X*_n_* film formed due to implantation is added to the fraction of the *i*-th surface θ_tCch_ covered with the M*_m_*X*_n_* compound film formed due to chemisorption. Formula (48) is written under the assumption that both films have the same sputtering coefficient, *S*_C_.

Further, the new model was supplemented with equations for reactive gas flows to all surfaces inside the vacuum chamber. The equations for the flow incident on the substrate (chamber wall), pumping out by a vacuum pump, and the balance of gas flows were written in the form (6), (14), and (15), respectively.

The flow incident on the target surface included both chemisorbed and implanted components:(49)$$Q_{t}=\left(2\alpha_{i}J_{X_{2}}\theta_{\mathrm{tM}}-\left(\frac{j}{e}\right)\beta \theta_{{\mathrm{tC}}_{\mathrm{ch}}}+2\left(\frac{j}{e}\right)f\right)A_{t}.$$

The first term in (49) describes the amount of gas consumed during chemisorption. The second term specifies the part of the flow from which atoms are implanted into the target as a result of the knock-on effect. Their contribution is subtracted because the model already includes these atoms in the first term. The last term in (49) takes into account the compound molecules formed in the bulk due to implantation.

The main result established in [119] and repeated in [122] is that the dependences *p* = *f*(*Q*_0_) obtained using the Berg chemisorption model and the RSD model are comparable. However, a difference was noted in the kinetics of gas flow during the first 10 s after turning on the magnetron. The Berg model predicts the transition from the metallic target operation mode to the oxide one with a shorter time constant. Calculations were performed for the sputtering of an aluminum target in Ar + O_2_.

### 4.2. Development of the RSD Model

In subsequent publications [65,120,121,122,123,124,125,126,127,128,129,130,131,132,133,134,135,136,137,138,139], Depla et al. reported on the development of the original RSD model. For example, in [65], kinetic equations appeared for all parts of the target surface $\partial \theta_{\mathrm{tM}}/\partial t,\;\partial \theta_{{\mathrm{tC}}_{\mathrm{ch}}}/\partial t\;\mathrm{and}\;\partial \theta_{\mathrm{tC}}/\partial t$ (see Figure 14), in which all considered physical processes are taken into account. Writing the kinetic equations in a different form did not change anything in the RSD model. This form was more common for writing equations describing the kinetics of processes on the target surface, which was adopted in describing the Berg model. However, a significant amendment has been made to them. If, in all previous cases, one of the main independent variables was the discharge current, then in [65], it was the ion current with the density:(50)$$j_{+}=-\frac{j}{\left(1+\gamma_{i}\right)},$$
where γ*_i_*, as in (3), is the ion-induced electron emission yield for the *i*-th part (*i* = M, C, C_ch_) of the target surface (as in (3)). As it follows from (50), the discharge current, including the electronic component, can be 5–10% higher than the ion current.

The authors of the RSD model paid quite a lot of attention to the simulation of the magnetron discharge [116,127,128,137]. To understand reactive sputtering, these problems are of independent importance. A change in the operating mode of the target, in particular, the transition from the metallic mode to the reactive mode and back, was reflected in *U = f*(*j*) and *U = f*(*Q*_0_). Jump-like changes appeared in them, corresponding to a change in the operating mode of the target, in which the ability of the surface to emit electrons under the influence of accelerated ions changed.

However, the simulation of the discharge, contributing to a better understanding of the processes of reactive sputtering, was not added to the previously considered models of new physical processes. Therefore, we leave aside the details of these publications and focus only on models that reflect the physicochemical aspects of reactive sputtering. We only note that the authors of these publications, when analyzing the current–voltage characteristics of magnetrons, consider the discharge voltage to be an independent variable. In the physics of a gas discharge, the discharge current is taken as this variable, which corresponds to the physical essence of the process of current flow through a gas.

The development of the RSD model is the subject of article [129], which presents a model of reactive sputtering of a cylindrical magnetron. In it, a cylindrical cathode rotated in a constant magnetic field. The authors of [129] noted that in the steady mode, the cylindrical magnetron behaved similarly to a planar magnetron. However, the change points of the target operation mode depended on the target rotation speed. It was found that the accepted form of the RSD model is insufficient to describe the reactive mode of operation of a rotating target. A more correct description of such a process was achieved by including the re-deposition of sputtered material on the target in the model. A similar correction for the target was proposed in [132,133]. A new modification of the RSD model described in [132,133] is called RSD2013. An interesting work is devoted to reactive high-power pulsed sputtering (R-HiPIMS), [135]. The difference between this method and DC sputtering or RF sputtering was the high proportion of ionization of the sputtered target material. As a result, the target sputtering rate increased. It turned out that in the R-HiPIMS process, the probability of observing hysteresis is much lower compared to DC magnetron sputtering. The reason for this could be related to the implantation of sputtered atoms into the target. However, the authors are right that for any reliable judgment about the features of R-HiPIMS, experimental data, which has been accumulated so far, is not enough.

In [129] and subsequent publications [130,132], the efforts of the authors were aimed at improving the computational procedures for solving problems of reactive sputtering modeling. For example, in [129], the surfaces of the target, substrate, and vacuum chamber are singled out to describe the process of reactive sputtering. Each of them is represented in the form of one or more cells. In this case, the model, as before, took into account the chemisorption of reactive gas molecules on the target, the implantation of chemisorbed particles due to the knock-on effect, and the direct implantation of reactive gas ions. It was pointed out that reactive gas atoms are implanted into the target volume, usually several nanometers below its surface. During direct ion implantation, molecular ions are neutralized immediately in front of the target surface, and upon deceleration in the target, they are split into two atoms. To describe the depth distribution of the concentration of compound molecules, the target was divided into 50 cells. In addition, to take into account the uneven flow of ions incident on the target surface, the sputtering area was divided into 10 × 25 cells.

Finally, let us turn our attention to publications devoted to the observation of double hysteresis in reactive sputtering [136,139]. This effect was observed in [139] when studying the dependence of the partial pressure of a reactive gas or discharge voltage on its flow. Under normal conditions, as follows from all the preceding, the resulting curve is S-shaped, indicating a possible hysteresis in the experiments. The authors of the mentioned works, under certain conditions, observed two S-shaped curves. One of them arose with an increase in the pressure of the reactive gas, the other—when returning to its original state due to a decrease in the pressure of the reactive gas. This behavior has been described as double hysteresis behavior. The origin of the double hysteresis behavior was studied by high-performance computing using a previously developed model. The influence of various process and material parameters has been evaluated on the basis of recently developed measures to characterize process design curves. This high throughput analysis showed that the double hysteresis behavior is related to the difference in the rate of removal of unreacted implanted ions as the reactive gas pressure is increased and decreased. In the parameter space, a region can be defined for which the double hysteresis behavior is strong. The latter can not only help in further experiments to study this behavior but also determine the conditions that limit its influence. It was found that for aluminum, a discharge current density of about 0.025 A/cm^2^ provides the maximum double hysteresis.

### 4.3. Implantation

The involvement of new physical processes in the model may become necessary if its initial version is inadequate. In 3.3, the inadequacy of the initial Berg model applied to the experimental results in [25,32] is shown. The reasons for the obtained result, as mentioned earlier, could be both the inaccuracy of the values of the model parameters that were used in the calculation and the incomplete understanding of the physical processes occurring during reactive sputtering.

The selection of model parameters is an optimization problem in which the minimum of a function is found in a space of several degrees of freedom. This function, in this case, serves as a criterion of adequacy when it is given by the quadratic form of the proximity between the experiment and the result of prediction by the model. An increase in the number of degrees of freedom, for example, in [130] there were five, significantly complicates the problem. However, if it is not possible to obtain an adequate model in this way, then one should look for additional physical processes or effects that affect the formation of a compound film on the target.

This was done in [75] for the initial Berg model when implantation and knock-on effect were added to it. However, in [119], the authors of the RSD model acted differently. Their initial model contained, as a mechanism for the formation of the M*_m_*X*_n_* compound on the target surface, the implantation of X atoms accompanied by a bulk chemical reaction and sputtering of the surface layers. The next steps in the development of the initial model were the inclusion of chemisorption and the knock-on effect in it. However, for the reader, the result was unexpected. Recall that the main result established in [119] was that the dependences *p* = *f*(*Q*_0_) obtained using the Berg chemisorption model and the RSD model were comparable. This is very strange since chemisorption and implantation should have an additive effect on the reactive sputtering process. In the RSD model, each process generates its own area on the target surface covered with the M*_m_*X*_n_* compound (see Figure 14).

In this regard, let us perform a more detailed analysis of the implantation of X atoms into the target. At the same time, let us pay attention to the result of the study given in [140], which the authors of the RSD model accepted as a reliable confirmation of the correctness of using implantation in the reactive sputtering model. This study was carried out by the method of nuclear reactions after turning off the magnetron. It was found that the maximum number of retained reactive gas atoms significantly exceeded the adsorbed monolayer. From our point of view, the physical reason for such a result could be different. This could be due, firstly, to polymolecular adsorption [141]. Second, the cause could be the diffusion of reactive gas atoms into the target volume. The significance of the second process can be expected to be quite high since, with a target thickness of 5–6 mm, the temperature in the region exposed to a powerful ion flux can reach 700–900 °C.

Let us discuss the primary RSD model based only on implantation. Initially, assuming that the formation of the M*_m_*X*_n_* film and its sputtering traditionally compete on the target surface, we write the kinetic equation for it in the form
(51)$$\frac{d\theta_{\mathrm{tC}}}{dt}={\left(\frac{d\theta_{\mathrm{tC}}}{dt}\right)}_{\mathrm{impl}}+{\left(\frac{d\theta_{\mathrm{tC}}}{dt}\right)}_{\mathrm{sput}},$$
where θ_tC_ is the fraction of the target surface covered by the compound (see Figure 14); (*d*θ_tC_/*dt*)_impl_ is the compound film growth rate due to ion implantation; (*d*θ_tC_/*dt*)_sput_ is the sputtering rate of the compound film sputtered by argon ions.

To write the first term in (51), we use the assumption of the authors of the RSD model that only a part of the reactive gas atoms α_i_ implanted in the volume takes part in the chemical reaction with the target metal atoms. Let us write this part of Equation (51) in the form adopted in [75]:(52)$${\left(\frac{d\theta_{\mathrm{tC}}}{dt}\right)}_{\mathrm{impl}}=2\alpha_{i}\left(\frac{j}{e}\right)f(1-\theta_{\mathrm{tC}})=2\alpha_{i}\left(\frac{j}{e}\right)\frac{p}{\left(p+p_{\mathrm{Ar}}\right)}(1-\theta_{\mathrm{tC}}).$$

The notation adopted in (52) was used in the previous sections of the paper.

Applying expression (52) in the model, the user encounters the first hard-to-resolve question: “What value of α_i_ should be taken in the calculations?”. The simplest α_i_ = 1 is unsuitable. The reason for this statement can be understood by performing a simple quantitative analysis.

Let us assume that the ion flux $X_{2}^{+}$ has a uniform spatial distribution, bombarding the entire surface of the target. Then the result of implantation can be presented in Figure 15, where the distribution of implanted atoms is shown with a gradient color, which has a maximum in the plane denoted by *x*_max_.

If α_i_ = 0, then in Equation (51) θ_tC_ ≡ 0, since the bulk chemical reaction does not occur in the target. Let us now assume that α_i_ = 1 and analyze the process of reactive sputtering of a silicon target in Ar + N_2_ at an ion energy of 0.45 keV, which was considered in [97]. In this case, participation of all implanted nitrogen atoms N in the chemical reaction is allowed. Then Figure 15 will reflect the change in the chemical composition of the nitrogen nitride film in the surface layer of the target. In dark areas, it can be close to the stoichiometric composition of Si_3_N_4_; in light areas, it will have nitrogen defects. Assuming the distribution of the concentration of nitrogen atoms N in silicon in the form of the Gaussian distribution (41), we determine the change in the relative concentration of implanted N atoms with time, taking sputtering into account, in the form of an integral with a variable upper limit:(53)$$\frac{c(x,\;t)}{D}=\frac{1}{\sqrt{2\pi }\Delta R_{p}}\int_{0}^{t}\exp\left[-\frac{{\left(x-a\tau -R_{p}\right)}^{2}}{2\Delta R_{p}^{2}}\right]d\tau ,$$
where *a* is the dimensional constant sputtering rate of the Si_3_N_4_ film. The result of integration (53) has the form
(54)$$\frac{c(x,\;t)}{D}=\frac{\mathrm{erf}\left(\frac{R_{p}+at-x}{\sqrt{2}\Delta R_{p}}\right)-\mathrm{erf}\left(\frac{R_{p}-x}{\sqrt{2}\Delta R_{p}}\right)}{2a}.$$

The parameters of the Gaussian distribution in (53) were determined using the SRIM program: *R_p_ =* 2.2 nm and Δ*R_p_ =* 1.1 nm. Figure 16 shows distribution (54) constructed with allowance for an arbitrarily chosen nitrogen nitride film sputtering rate (*dx*/*dt*)_sput_ = *a* = 0.2 nm/s.

Despite the fact that the ion with the energy of 0.45 keV was represented in the molecular form $N_{2}^{+}$, in SRIM calculations, Figure 16 describes the change in the concentration of nitrogen atoms N implanted in the target. This remark is based on the assumptions of the RSD model. Recall that it assumes deceleration in the target of an ion that dissociates into two atoms. If an atomic nitrogen ion is set as an ionized particle, then its projected range *R_p_* will be equal to 28 nm at Δ*R_p_* = 1.6 nm.

The surface concentration of atoms implanted into the target, taking into account sputtering, as Figure 16 demonstrates, will increase. Its change is shown in Figure 17; after 40 s, the process will enter the steady mode when the value *c*(0, *t*)/*D* reaches its maximum value. This can only mean one thing, the target cannot have a metallic mode. After turning on the magnetron, a compound film will be formed on the entire surface of the target, the chemical composition of which will continuously change, tending towards the stoichiometric one. In this case, the film thickness will also remain unchanged in Equation (51) θ_tC_ ≡ 1 since all implanted nitrogen atoms participate in the bulk chemical reaction.

From all that has been said, it follows that the inequality α_i_ < 1 is valid in (52), and the final result largely depends on the choice of α_i_.

The results presented by the authors of the RSD model are interesting, but they do not make it possible to assess the state of the target. For example, it does not matter at all why there was no jump in the dependence θ_tCs_ = *f* (*f*) and why the model result differed from the experimental one in [97]. The main thing, at first glance, is the ambiguity of how to use the obtained result for modeling since it does not provide information about the main function of the reactive sputtering process *p* = *f* (*Q*_0_), which was previously accepted in [44,45] and many other publications (see Section 3.1). In [119], it was not possible to find an answer to the question of how the functions θ_tC_ = *f* (*f*) and θ_tCs_ = *f* (*f*) can be used to determine the boundary between the metallic and reactive modes of target operation.

## 5. Nonisothermal Physicochemical Model (the Barybin Model)

Let us return to the initial isothermal Berg model in [46] based on chemisorption. Chemisorption under conditions of constant temperature is used in all models considered in this work. Although in [65], the authors noted that the Berg model uses “…crude approximations…”. However, they did not abandon this model.

Section 3.3 of this paper notes two very strong simplifications of the Berg model. They are the replacement of the chemical reaction of M*_m_*O*_n_* film formation by the chemisorption mechanism and the assumption that the temperatures of all surfaces are equal.

For the last twenty years at Saint Petersburg Electrotechnical University (LETI, Russia), our group has been studying transition metal oxide, nitride, and oxynitride films deposited by reactive magnetron sputtering [142,143,144,145,146,147,148,149]. Initial interest in a traditional magnetron with a well-cooled metal target (cold target) made it possible to create a sputter assembly with a hot target [150,151,152,153] and then with a sandwich target [154,155,156,157,158,159]. The study of the processes that occur during film deposition using these tools inevitably led to modeling [160,161,162,163,164,165]. The starting point in this work was the article [46]. Developing the main assumptions of the Berg model outlined earlier, we proposed to bring the model closer to reality. Firstly, chemisorption was replaced by chemical reactions, and secondly, the limitation on surface temperatures inside the vacuum chamber was removed. The model was named “nonisothermal physicochemical” or the Barybin model after the name of the group leader in the first decade of the 21st century.

### 5.1. Single Cold Target in Ar + X_2_

The first model was developed to study the process of reactive sputtering of a single cold target in an Ar + X_2_ mixture [160]. It adopted the following assumptions:The vacuum chamber contains a target, a substrate, and a wall. The areas of these elements are equal to *A*_t_, *A*_s_, and *A*_w_ (or *A_i_*, *i* = t, s, w), respectively;During film synthesis, the surface temperatures *T*_t_, *T*_s_, and *T*_w_ (or *T_i_*, *i* = t, s, w) are different, and the temperature of the gas environment *T*_0_ is equal to the wall temperature *T*_w_;Every surface can undergo a chemical reaction:
(55)$$M+\frac{n}{2m}X_{2}\overset{k(T_{i})}{\to }\frac{1}{m}M_{m}X_{n},\;i=t,\;s,\;w.$$

Surface reaction (55) proceeds between the adsorbed X_2_ reactive gas molecules and the metal on the target surface fractions free from M*_m_*X*_n_*. In (55), the quantity *k*(*T_i_*) is the Arrhenius reaction rate constant, which has the dimension of the flux density:(56)$$k\left(T_{i}\right)=k_{0}\exp\left(-\frac{E_{a}}{kT_{i}}\right),$$
where *k*_0_ and *E_a_* are the constant and activation energy of the reaction, respectively.

4.At any time (see Figure 2):Relative fraction θ_t_ of the sputtered target surface is covered with M*_m_*X*_n_*. The rest (1 − θ_t_) is pure metal M. This state can arise due to the competition of two processes, including the formation of the M*_m_*X*_n_* film according to surface reaction (55) and its sputtering with argon ions in the form of molecules. The flux sputtered from the target surface includes *J*_tM_ (pure metal atoms M) and *J*_tC_ (M*_m_*X*_n_* molecules);Due to the surface reaction (55) and fluxes generated by the target, a solid solution M + M*_m_*X*_n_* is formed on the *i*-th surface (*i* = s, w). Let us represent it on each surface as two regions with relative areas θ*_i_* and 1 − θ*_i_*, containing the compound M*_m_*X*_n_* and the metal M, respectively.5.Each surface consumes reactive gas to maintain surface reaction (55). Let *Q_i_* denote the flux incident on the *i-*th surface (*i* = t, s, w) and participating in the formation of the M*_m_*X*_n_* compound on it.

The independent variables in this problem are the discharge current density *j* and the reactive gas flow *Q*_0_ introduced into the chamber. The main dependent variable of the process is the partial pressure *p* of the gas X_2_.

Following [46], we will compose a system of algebraic equations describing the kinetics of processes occurring on all surfaces and gas flows inside the vacuum chamber.

The kinetic equation for the target surface, by analogy with (9), takes the form
(57)$$\frac{d\theta_{t}}{dt}\;\;=\;\;{\left(\frac{d\theta_{t}}{dt}\right)}_{\mathrm{ch}}+\;\;{\left(\frac{d\theta_{t}}{dt}\right)}_{\mathrm{sp}}.$$

However, unlike (10), the first term in (57) describes the formation of the M*_m_*X*_n_* film due to the surface chemical reaction (55). In accordance with the law of mass action [166], we set the reaction rate with the expression
(58)$${\left(\frac{d\theta_{t}}{dt}\right)}_{\mathrm{ch}}=\frac{1}{N_{\mathrm{ch}}}k(T_{t})\;\theta_{0t}^{n/2m}(1-\theta_{t}),$$
here *N*_ch_ is the concentration of chemical reaction centers on the target surface; θ_0t_ is the fraction of the target surface covered with adsorbed gas molecules. Physical adsorption is described by the Langmuir isotherm, which for nonisotemic conditions can be represented as:(59)$$\theta_{0t}(T_{0},T_{t},p)=\frac{b(T_{0},T_{t})p}{1+b(T_{0},T_{t})p}.$$

In (59), parameter *b*(*T*_0_, *T*_t_) is the function of gas temperature *T*_0_, and target temperature *T*_t_:(60)$$b(T_{0},T_{t})=\frac{\alpha_{0}\tau_{a}}{N_{\mathrm{ph}}\sqrt{2\pi m_{0}kT_{0}}}\exp\left(\frac{Q_{\mathrm{ph}}}{RT_{t}}\right),$$
where α_0_ is the condensation coefficient of reactive gas molecules; *N*_ph_ is the concentration of physical adsorption centers τ_a_ ≈ 10^−13^ s is the average lifetime of a molecule in the adsorbed state; *Q*_ph_ is the heat of physical adsorption (cal/mol); *R* = 1.99 cal/(mol· K) is the universal gas constant.

With the second term in (57) (*d*θ_t_/*dt*)_sp_ = −*jS*_C_θ_t_/*eN*_ch_, the steady state equation for the target surface takes the form
(61)$$k(T_{t})\;\theta_{0t}^{n/2m}(1-\theta_{t})=\;\;\left(\frac{J}{e}\right)S_{C}\theta_{t},$$

Using (13), the kinetic equation of the steady state of the wall and substrate surfaces can be written as:(62)$$\begin{array}{l} k(T_{i})\;\theta_{0i}^{n/2m}(1-\theta_{i})+\;\left(\;\frac{j}{e}\right)\frac{A_{t}}{A_{w}+A_{s}}\;S_{C}\theta_{t}\left(1-\theta_{i}\right)=\\ \;\;\;\;\;\;\;\;\;\;\;\;\;\;\;=\;\;\left(\frac{j}{e}\right)\frac{A_{t}}{A_{w}+A_{s}}\;\frac{S_{M}}{\;m}\left(1-\theta_{t}\right)\;\theta_{i}\; \end{array},\;i=w,\;s.$$

In (62), θ_0*i*_ is given by Equations (59) and (60), in which *T*_t_ is replaced by *T_i_*, *i* = w, s.

The gas flows supporting the chemical reaction on all surfaces are described by the expression:(63)$$Q_{i}=\;\frac{n}{2}c^{0}k(T_{i})\theta_{0i}^{n/2m}(1-\theta_{i})A_{i},\;i=t,\;w,\;s,$$
with the dimensional factor *c*^0^ = 2.436 × 10^−18^, (cm^3^×s)/min, which helps to convert the gas flow from the number of particles per second to cubic centimeters per minute (sccm), standard for technological tasks. When calculating *c*_0_, as usual, *T*^0^ = 298 K and *p*^0^ =10^5^ Pa were chosen as standard conditions. Denoting the volume of one sputtered particle per one minute through *V*_1_, we obtain:$$c^{0}=60V_{1}=60\frac{p^{0}}{kT^{0}}=2.436\times 10^{-18},\;\frac{{\mathrm{cm}}^{3}\times s}{\min}.$$

The balance equations for gas flows
(64)$$Q_{0}=Q_{t}+Q_{s}+Q_{w}+Q_{p}$$
and pumping out close the system:(65)$$Q_{p}=cpS_{p},$$
where *S*_p_ is the chamber pumping speed (m^3^/s); *c* = 600, (s×cm^3^)/(Pa×m^3^×min) is the dimensional factor that converts pascal per cubic meter per second to cubic centimeters per minute under standard conditions.

The system of 8 algebraic Equations (61)–(65), taking into account (59) and (60) with respect to the function *p* = *f* (*Q*_0_, *j*), should be solved numerically, determining the one-dimensional dependences *p* = *f* (*Q*_0_) for *j* = const or *p* = *f* (*j*) for *Q*_0_ = const. The initial data for calculations are the values of the following parameters: *j*, α_0_, τ_0_, *S*_M_, *S*_C_, *S*_p_, *A*_t_, *A*_s_, *A*_w_, *N*_ph_, *Q*_ph_, *T*_t_, *T*_s_, *T*_w_, *E_a_*, *k*_0_. The model makes it possible to estimate the influence of independent variables on the values θ_t_, θ_w_, θ_s_, *Q*_t_, *Q*_w_, *Q*_s_, and *Q*_p_. Direct measurement of these variables is not possible.

The proposed model was used to simulate the reactive sputtering of a tantalum target. The experimental results of the process are taken from [32]. The result of the numerical solution of Equations (61)–(65) with allowance for (59) and (60) at a current density *j* = 0.05 A/cm^2^ and parameters from Table 1 is shown in Figure 4 of [160]. This result shows at a qualitative level that the Barybin nonisothermal physicochemical model of reactive sputtering can adequately describe the experimental results.

### 5.2. Single Cold Target in Ar + O_2_ + N_2_

Films of transition metal oxynitrides with composition expressed by the chemical formula M*_m_*O*_n_*N*_p_* in many cases form a continuous series of compounds with different concentrations of nitrogen and oxygen (from M*_m_*O*_n_* oxide to M*_m_*N*_p_* nitride) [167].

For the deposition of oxynitride films, the method of DC reactive magnetron sputtering is largely used. This method is most often based on the sputtering of a metal target in a mixture of argon, oxygen, and nitrogen [168,169,170,171,172,173,174]. Let us apply the nonisothermal physicochemical model for such processes. The main assumptions 1 and 2 of this model, set out in Section 5.1, are also valid in this case. Assumptions 3, 4, and 5 should be changed. In this regard, the physical model of this process will be expressed by a number of the following assumptions.

The vacuum chamber contains a target, a substrate, and a wall with surface areas *A_i_*, *i* = t, s, w.The temperatures of the surfaces *T_i_*, *i* = t, s, w are different, and the temperature of the gas environment *T*_0_ is equal to the wall temperature *T*_w_.Metal oxynitride appears as a result of a surface chemical reaction on *A*_t_, *A*_s_, and *A*_w_ surfaces. We assume that this compound arises in the form of a solid solution consisting of M*_m_*O*_n_* oxide and MN nitride. In this case, the components of the solution are formed due to the occurrence of two independent reactions:
(66)$$M+\frac{n}{2m}O_{2}\overset{k_{1}(T)}{\to }\frac{1}{m}M_{m}O_{n},$$
(67)$$M+\frac{1}{2}N_{2}\overset{k_{2}(T)}{\to }\mathrm{MN},$$
were *k*_1_(*T*) and *k*_2_(*T*) are the reaction rate constants. Here and below, index 1 will correspond to oxide or oxygen, and index 2 to nitride or nitrogen. The accepted chemical formula for nitride in the form of MN corresponds to many compounds of this type (TiN, AlN, TaN, etc.).At any time:Relative parts θ_t_1__ and θ_t_2__ of the sputtered target surface are covered by M*_m_*O*_n_* oxide and MN nitride, respectively (Figure 18). The rest (1 − θ_t_1__ − θ_t_2__) is pure metal M. This state can arise due to the competition of two processes, including the formation of the M*_m_*O*_n_* + MN solid solution film according to reactions (66) and (67) and its sputtering by argon ions in the form of molecules. The flux sputtered from the target surface includes *J*_tM_ (pure metal atoms M), *J*_tC_1__ (M*_m_*O*_n_* molecules), and *J*_tC_2__ (MN molecules);Due to surface reactions (66) and (67) and fluxes generated by the target, a solid solution M + M*_m_*O*_n_* + MN is formed on the *i*-th surface (*i* = s, w). Let us represent it on each surface as three regions with relative areas θ*_i_*1*__*, θ*_i_*2*__*, and 1 − θ*_i_*1*__* − θ*_i_*2*__* containing oxide M*_m_*O*_n_*, nitride MN, and metal M, respectively.Each surface consumes reactive gas to maintain surface reactions (66) and (67). Let *Q_i_j__* denote the fluxes incident on the *i-*th surface (*i* = t, s, w) and participating in the formation of oxide M*_m_*O*_n_* (*j* = 1) and nitride MN (*j* = 2).

The analytical description of the physical model (items 1–5) is expressed, as in Section 5.1, by a system of equations. The solution of this system will determine the relationship between the dependent and independent variables of the reactive sputtering process. In this case, there are three independent variables: the current density of argon ions on the target *j* and the incoming gas flows *Q*_0_1__ and *Q*_0_2__. The main dependent variables of the process are the partial pressures of oxygen *p*_1_ and nitrogen *p*_2_. From the further presentation, it will become clear that the process is characterized by several more dependent variables, namely: *Q*_t_1__, *Q*_s_1__, *Q*_w_1__, *Q*_t_2__, *Q*_s_2__, and *Q*_w_2__.

Let us consider the process in more detail and derive equations for finding the dependencies *p*_1_ = *f* (*j*, *Q*_0_1__, *Q*_0_2__) and *p*_2_ = *f* (*j*, *Q*_0_1__, *Q*_0_2__).

The state of the target surface for this process, in contrast to (56), must be described by two kinetic equations:(68)$$\frac{d\theta_{t_{j}}}{dt}\;\;=\;\;{\left(\frac{d\theta_{t_{j}}}{dt}\right)}_{\mathrm{ch}}+\;\;{\left(\frac{d\theta_{t_{j}}}{dt}\right)}_{\mathrm{sp}},\;j=1,\;2.$$

The kinetics of the compound formation on the target surface in (68) is given by two equations:(69)$${\left(\frac{d\theta_{t_{1}}}{dt}\right)}_{\mathrm{ch}}=\;\frac{k_{1}(T_{t})}{N_{{\mathrm{ch}}_{1}}}\;\theta_{{0t}_{1}}^{n/2m}(1-\theta_{t_{1}}-\theta_{t_{2}}),$$
(70)$${\left(\frac{d\theta_{t_{{}_{2}}}}{dt}\right)}_{\mathrm{ch}}=\;\frac{k_{2}(T_{t})}{N_{{\mathrm{ch}}_{2}}}\;\theta_{{0t}_{2}}^{1/2}(1-\theta_{t_{1}}-\theta_{t_{2}}),$$
where *N*_ch_*j*__ (*j* = 1, 2) are the concentrations of the chemical reaction centers on the target surface; θ_0t_*j*__ (*j* = 1, 2) are the fractions of the target surface covered by the adsorbed molecules of oxygen and nitrogen. Let us describe physical adsorption in the form of the Langmuir isotherm by analogy with (59):(71)$$\theta_{0t_{j}}(T_{0},\;T_{t},\;p_{j})=\frac{b_{j}(T_{0},\;T_{t})p_{j}}{1+b_{1}(T_{0},\;T_{t})p_{1}+b_{2}(T_{0},\;T_{t})p_{2}},\;j=1,\;2.$$

In (71), parameter *b*(*T*_0_, *T*_t_) is the function of gas temperature *T*_0_, and target temperature *T*_t_:(72)$$b_{j}(T_{0},\;\;T_{t})=\frac{\alpha_{0}\tau_{a}}{N_{ph}\sqrt{2\pi m_{0_{j}}kT_{0}}}\exp\left(\frac{Q_{{\mathrm{ph}}_{j}}}{RT_{t}}\right),\;j=1,\;2,$$
where *Q*_ph_*j*__ is heat of physical adsorption of the *j*-th gas. The remaining designations fully correspond to Equation (60).

The second terms in (68) are obvious; therefore, following (21) and (22), we write the equations for the steady state of the target surface at the sputtering yields of oxide *S*_C_1__ and nitride *S*_C_2__ in the form
(73)$$k_{1}(T_{t})\;\theta_{{0t}_{1}}^{n/2m}(1-\theta_{t_{1}}-\theta_{t_{2}})=\;\;\left(\frac{J}{e}\right)S_{C_{1}}\theta_{t_{1}},$$
(74)$$k_{2}(T_{t})\;\theta_{{0t}_{2}}^{1/2}(1-\theta_{t_{1}}-\theta_{t_{2}})=\;\;\left(\frac{J}{e}\right)S_{C_{2}}\theta_{t_{2}},$$

Further, using (62), the kinetic equations for the steady state of the wall and substrate surfaces can be expressed in the form
(75)$$\begin{array}{l} k_{1}(T_{i})\theta_{0i_{1}}^{n/2m}(1-\theta_{i_{1}}-\theta_{i_{2}})+\left(\frac{J}{e}\right)\frac{A_{t}}{A_{w}+A_{s}}[S_{C_{1}}\theta_{t_{1}}(1-\theta_{i_{1}}-\theta_{i_{2}})+\\ +S_{C_{1}}\theta_{t_{1}}\theta_{i_{2}}-\frac{S_{M}}{m}(1-\theta_{t_{1}}-\theta_{t_{2}})\theta_{i_{1}}-S_{C_{2}}\theta_{t_{2}}\theta_{i_{1}}]=0,\;\;\;i=s,\;\;w, \end{array}$$
(76)$$\begin{array}{l} k_{2}(T_{i})\theta_{0i_{2}}^{1/2}(1-\theta_{i_{1}}-\theta_{i_{2}})+\left(\frac{J}{e}\right)\frac{A_{t}}{A_{w}+A_{s}}[S_{C_{2}}\theta_{t_{2}}(1-\theta_{i_{1}}-\theta_{i_{2}})+\\ +S_{C_{2}}\theta_{t_{2}}\theta_{i_{1}}-S_{M}(1-\theta_{t_{1}}-\theta_{t_{2}})\theta_{i_{2}}-S_{C_{1}}\theta_{t_{1}}\theta_{i_{2}}]=0,\;\;\;i=s,\;\;w. \end{array}$$

In (75) and (76) θ_0*i_j_*_ (*j =* 1, 2) are given by Equations (71) and (72), where *T*_t_ is replaced by *T_i_*, *i* = w, s.

The incoming reactive gas flows *Q*_0_1__ and *Q*_0_2__ have a drain on all surfaces:Pumped gas flows *Q*_p_1__ and *Q*_p_2__, which for *j* =1, 2, are written by analogy with Equation (64):
(77)$$Q_{p_{j}}=cp_{j}S_{p};$$Flows to the target *Q*_t_1__ and *Q*_t_2__;Flows to the substrate, where the compound film is formed from the target material, *Q*_s_1__ and *Q*_s_2__;Flows to the chamber internal walls, where the target material is deposited, *Q*_w_1__ and *Q*_w_2__.

By analogy with (62), for the flows of oxygen and nitrogen to the *i*-th surface (*i* = t, s, w), we can write:(78)$$Q_{i_{1}}=\frac{n}{2}c^{0}k_{1}(T_{i})\theta_{0i_{1}}^{n/2m}(1-\theta_{i_{1}}-\theta_{i_{2}})A_{i};$$
(79)$$Q_{i_{2}}=\frac{1}{2}c^{0}k_{2}(T_{i})\theta_{0i_{2}}^{1/2}(1-\theta_{i_{1}}-\theta_{i_{2}})A_{i}.$$

Now the balance equations of the gas flows (*j =* 1, 2) take the form
(80)$$Q_{0_{j}}=Q_{t_{j}}+Q_{s_{j}}+Q_{w_{j}}+Q_{p_{j}}.$$

The balance equations for gas flows (80) at *j* = 1, 2 close the system of equations describing the reactive sputtering process under study. The system, taking into account (71) and (72), includes 16 Equations (73)–(80) with the values of subscripts *i* = t, s, w and *j* = 1, 2.

In particular cases, at *Q*_0_1__ = 0 or *Q*_0_2__ = 0, the resulting system of equations describes the process of sputtering a metal target in an environment of one reactive gas (nitrogen or oxygen). It is easy to show that, in these cases, the system (73)–(80) reduces to the system of Equations (61)–(65). To do this, it is necessary to substitute O_2_ or N_2_ as the reactive gas X_2_ into the equation of the surface chemical reaction (55).

The system of Equations (73)–(80) was applied to simulate the reactive sputtering of a titanium target in an Ar + O_2_ + N_2_ gas mixture. The process parameters are given in Table 2.

The results of numerical solution of Equations (73)–(80), taking into account (71) and (72) at the current density of *j* = 200 A/m^2^, are shown in Figure 19.

Figure 19 shows that the proposed model makes it possible to predict the points of change in the operating mode of the target, indicating the hysteresis effect. With an increase in the incoming nitrogen flow, both points of change in the operating mode of the target shift to the region of lower values of the incoming oxygen flow, which is quite expected. In this case, the form of dependences *p*_1_ = *f*(*Q*_0_1__) changes with increasing *Q*_0_2__.

### 5.3. Single Hot Target in Ar + X_2_

In recent years, the hot-target magnetron has gained increased attention among specialists [150,151,152,155,156,157,177,178,179,180,181,182]. A feature of this device is that its target can be heated to melting. In a heated state, the target becomes a source of thermoelectrons and evaporated particles. These two fluxes change the conditions of reactive sputtering, so they must be taken into account in the physical model of the process [162,164,165].

Let us compose a model of reactive sputtering of a single hot target. As its basis, by analogy with Section 5.1, we will take several assumptions.

The vacuum chamber contains a target, a substrate, and a wall with surface areas *A_i_*, *i* = t, s, w.The temperatures of the surfaces *T_i_*, *i* = t, s, w are different, and the temperature of the gas environment is *T*_0_ = *T*_w_. The target surface temperature *T*_t_ depends on the current density *j*.On a part of each surface free from the M*_m_*X*_n_* film, chemical reaction (55) can occur, which is preceded by the physical adsorption of molecules of reactive gas X_2_.At any moment in time (see Figure 20):The relative fraction θ_t_ of the sputtered target surface, as in any similar cases, is covered by M*_m_*X*_n_*. The rest (1 − θ_t_) is a pure metal M. The flux sputtered from the target surface includes fluxes *J*_tM_ (atoms of pure metal M) and *J*_tC_ (molecules M*_m_*X*_n_*)
(81)$$J_{\mathrm{tM}}(j_{+})=J_{\mathrm{tM}}^{\mathrm{sp}}(j_{+})+J_{\mathrm{tM}}^{\mathrm{ev}}\left[T_{t}(j_{+})\right];$$
(82)$$J_{\mathrm{tC}}(j_{+})=J_{\mathrm{tC}}^{\mathrm{sp}}(j_{+})+J_{\mathrm{tC}}^{\mathrm{ev}}\left[T_{t}(j_{+})\right].$$The fluxes of sputtered metal atoms and compound molecules with densities $J_{\mathrm{tM}}^{\mathrm{sp}}(j_{+})$ and $J_{\mathrm{tC}}^{\mathrm{sp}}(j_{+})$ are proportional to the ion current density *j*_+_. The corresponding fluxes of evaporated particles with densities $J_{\mathrm{tM}}^{\mathrm{ev}}\left[T_{t}(j_{+})\right]$ and $J_{\mathrm{tC}}^{\mathrm{ev}}\left[T_{t}(j_{+})\right]$ depend indirectly on *j*:(83)$$J_{\mathrm{tM}}^{\mathrm{ev}}\left[T_{t}(j_{+})\right]=\frac{10^{\left(A_{M}-\frac{B_{M}}{T_{t}(j_{+})}\right)}}{\sqrt{2\pi m_{M}kT_{t}(j_{+})}}.$$
(84)$$J_{\mathrm{tC}}^{\mathrm{ev}}\left[T_{t}(j_{+})\right]=\frac{10^{\left(A_{C}-\frac{B_{C}}{T_{t}(j_{+})}\right)}}{\sqrt{2\pi m_{C}kT_{t}(j_{+})}}.$$
where *A*_M_, *B*_M_, *A*_C_, and *B*_C_ are constants that determine the saturated vapor pressure of the metal M and the compound M*_m_*X*_n_* in pascals, respectively; *m*_M_ and *m*_C_ are the masses of the metal atom M and the M*_m_*X*_n_* molecule, respectively;On the *i*-th surface (*i* = s, w), due to the surface reaction (55) and fluxes generated by the target, a solid solution M + M*_m_*X*_n_* is formed. Let us represent it on each surface as two regions with relative areas θ*_i_* and 1 − θ*_i_* containing the compound M*_m_*X*_n_* and the metal M, respectively (see Figure 20).A flow of reactive gas *Q_i_* falls on each surface, supporting the formation of the compound M*_m_*X*_n_* according to reaction (55).The discharge current density is
(85)$$j=(1+\gamma )\;j_{+}+j_{-},$$
where γ is the potential ion-induced secondary electron emission yield; *j_+_* is the ion current density; *j*_−_ is the current density of thermal electron emission (described by the Richardson–Dushman equation):(86)$$j_{-}(T_{t})=AT_{t}^{2}\exp\left(-\frac{\phi_{t}}{kT_{t}}\right),$$
where *A* ≈ 120 A∙cm^−2^∙K^−2^; φ_t_ is the work function of electrons for the target material; *k* is the Boltzmann constant.The influence of the temperature on the parameters of the sputtering process (sputtering yield and ion-induced electron emission yield, etc.) is insignificant.

The independent variables in this problem, as before, are the discharge current density *j* and the reactive gas flow *Q*_0_ introduced into the chamber. The main dependent variable of the process is the partial pressure *p* of the gas X_2_.

Let us describe the physical model of reactive sputtering, expressed by assumptions 1–7, by a system of equations. It, as in all cases, should include two groups of equations that describe the steady state of three surfaces and gas flows in the vacuum chamber, respectively.

The kinetic equation for the target surface has the form:(87)$$\frac{d\theta_{t}}{dt}\;\;=\;\;{\left(\frac{d\theta_{t}}{dt}\right)}_{\mathrm{ch}}+\;\;{\left(\frac{d\theta_{t}}{dt}\right)}_{\mathrm{sp}}+{\left(\frac{d\theta_{t}}{dt}\right)}_{\mathrm{ev}}.$$

The first term in (87) describes the formation of the M*_m_*X*_n_* film according to reaction (55) and has the form of (58). Let us repeat here (58) for the convenience of readers:(88)$${\left(\frac{d\theta_{t}}{dt}\right)}_{\mathrm{ch}}=\;\;\frac{k(T_{t}\left(j\right))}{N_{\mathrm{ch}}}\;\theta_{0t}^{n/2m}(1-\theta_{t}).$$

In Equation (88), θ_0t_ is given by Equations (59) and (60). The second term in (87) equals
(89)$${\left(\frac{d\theta_{t}}{dt}\right)}_{\mathrm{sp}}=-\frac{1}{N_{\mathrm{ch}}}\frac{j-j_{C}^{-}(T_{t})}{e(1+\gamma_{C})}S_{C}\theta_{t},$$
where for the compound M*_m_*X*_n_*: γ_C_ is the potential ion-induced secondary electron emission yield; *S*_C_ is the sputtering yield; $j_{C}^{-}(T_{t})$ is the current density of thermal electron emission defined by Equation (86). The factor $\left[j-j_{C}^{-}(T_{t})\right]/\left[1+\gamma_{C}\right]$ in the right part of (89) sets the density of ion current on the target. The third term in (87) is defined by the Hertz–Knudsen equation in the form [183]
(90)$${\left(\frac{d\theta_{t}}{dt}\right)}_{\mathrm{ev}}=-\frac{1}{N_{\mathrm{ch}}}\frac{10^{\left(A_{C}-\frac{B_{C}}{T_{t}}\right)}}{\sqrt{2\pi m_{C}kT_{t}}}\theta_{t},$$

Substituting (88), (89), and (90) to Equation (87), we can write the kinetic equation for the target steady state (*d*θ_t_/*dt* = 0):(91)$$k(T_{t})\;\theta_{0}^{n/2m}(1-\theta_{t})=\;\;J_{\mathrm{tC}}\theta_{t}.$$

Let us express the compound flux density *J*_tC_ in (91) in more detail using (82), (89), and (90):(92)$$J_{\mathrm{tC}}=\;\;\left(\frac{j-j_{C}^{-}(T_{t})}{e(1+\gamma_{C})}S_{C}+\frac{10^{\left(A_{C}-\frac{B_{C}}{T_{t}}\right)}}{\sqrt{2\pi m_{C}kT_{t}}}\right).$$

The steady state equation for the surface of the substrate and the wall during sputtering of a hot target has a form that is similar to Equation (62). Figure 20 shows its difference; that is, when describing the processes on these surfaces, it is necessary to take into account the fluxes of evaporated particles. Therefore, we write (62) in the form
(93)$$k(T_{i})\;\theta_{0}^{n/2m}(1-\theta_{i})\;+\;\frac{A_{t}}{A_{s}+A_{w}}\;J_{\mathrm{tC}}\theta_{t}\left(1-\theta_{i}\right)=\frac{A_{t}}{A_{s}+A_{w}}J_{\mathrm{tM}}\left(1-\theta_{t}\right)\theta_{i},\;i=s,\;w.$$

In (93), *J*_tM_ is written by analogy with (92):(94)$$J_{\mathrm{tM}}=\left(\frac{j-j_{M}^{-}(T_{t})}{e(1+\gamma_{M})}S_{M}+\frac{10^{\left(A_{M}-\frac{B_{M}}{T_{t}}\right)}}{\sqrt{2\pi m_{M}kT_{t}}}\right),$$
where for metal M: γ_M_ is the potential ion-induced secondary electron emission yield; $j_{M}^{-}(T_{t})$ is the current density of thermal electron emission; *S*_M_ is the sputtering yield.

The second group of equations describes:Reactive gas flows on each of the three internal surfaces of the vacuum chamber (63);Balance of gas flows (64);Pumping out the vacuum chamber (65).

The system of algebraic equations describing the model of reactive sputtering of a hot target includes Equations (63)–(65), (91) and (93). If target heating initiates insignificant thermal electron emission and evaporation, then the result of solving system (63)–(65), (91) and (93) will not differ from that predicted by the reactive sputtering model of a cold target (61)–(65) taking into account (59) and (60).

As an example, the system (63)–(65), (91) and (93) was used in modeling reactive sputtering of a hot titanium target in an Ar + N_2_ mixture [165]. The solution is fulfilled for the function *p* = *f* (*j*, *Q*_0_). The chemical reaction in this problem is described by Equation (63). In the calculations, the parameters indicated in Table 3 were used.

The measurement of the target temperature is of great importance. Most often, non-contact infrared pyrometric methods are used in this case [189,190,191]. In this work, the dependence of the target temperature on the discharge current density was determined by a technique based on measuring the optical spectra of the discharge and separating from them a component associated with the thermal radiation of a heated target [192,193]. The results of this computational procedure are shown with dots in Figure 21.
(95)$$T_{t}\approx 2000-1440e^{-0.00239j}.$$

Next, we perform an analysis of the features of the hot target sputtering process, which were revealed by the proposed model. For comparison, a similar problem was solved for a cold target.

An analysis of the dependences *p* = *f* (*j*, *Q*_0_) obtained using the proposed model (given in [165]) showed that for a hot target:Hysteresis persists over a wide temperature range. In accordance with expression (95), current densities of 50 and 500 A/m^2^ corresponded to temperatures of 720 and 1560 K;Changes in the operating mode of the target occur at lower values of *Q*_1_ and *Q*_2_ compared to a cold target (Figure 22);The width of the hysteresis loop also has smaller values (Figure 23).

The results of a more detailed analysis are shown in Figure 23, which presents the dependences Δ*Q* = *Q*_1_ − *Q*_2_ = *f* (*j*) for cold (CT curve) and hot (HT curve) targets. In addition, Figure 23 shows a similar dependence (dashed curve) obtained by solving Equations (63)–(65), (91) and (93), from which the terms taking into account evaporation were removed. Figure 23 demonstrates that at *j* < 400 A/m^2^ (corresponding to a target temperature of 1450 K), evaporation does not influence the sputtering process.

The presented results confirm that the proposed model can predict the reactive sputtering process of a single hot target.

In [163], we published the results of modeling the reactive sputtering process of a single hot target in a gas mixture of Ar + O_2_ + N_2_. The equations describing this model are similar to (73)–(80), taking into account the features of sputtering of a hot target expressed by formulas (81)–(86). We will not delve into the details of this model. Let us present only some results of calculations for the model of hot titanium target sputtering.

The system of equations describing the model was solved for incoming oxygen flows *Q*_01_ from 0 to 10 sccm, constant nitrogen flows *Q*_02_ 0.1, 0.2, and 0.3 sccm, and current densities from 200 to 400 A/m^2^. All dependencies, *p = f*(*Q*_01_), were S-shaped, characteristic of the hysteresis effect. For comparison, solid lines in Figure 24 show the dependencies *p* = *f* (*Q*_01_)*_Q_*_02 = const_, calculated from the cold target sputtering equations (see expressions (73)–(80)). It can be seen from Figure 24 that when a hot target is sputtered, the addition of nitrogen shifts to the left the points of change in the operating modes of the target, at which the derivative *dp*/*dQ*_01_ undergoes discontinuities. In addition, an increase in nitrogen incoming flow leads to a decrease in the width of the hysteresis region.

### 5.4. Sandwich Target in Ar + X_2_

The sputtering unit of the magnetron containing several metal plates with cut-outs on the same axis is called the “sandwich target”. Let us construct a physical model for reactive sputtering of the simplest sandwich target with two plates using several assumptions [194].

The vacuum chamber includes four elements: the internal and external plates located in the sputtering unit on the same axis [194], the substrate, and the wall. The areas of these elements are denoted by *A_i_*, *i* = t_1_, t_2_, s, w. Cut-outs with a total area of *A*_co_ are made in the external plate. The values *A*_t_1__ and *A*_t_2__ specify the areas of only the sputtered fractions of the plates. On the external plate, this fraction is ring-shaped with geometric dimensions determined by the magnetic field. If the area of the ring is *A*, then the actual sputtered area of the external plate is *A*_t_2__ = *A* − *A*_co_ or in relative units *A*_t_2__/*A* = (*A* − *A*_co_)/*A* = 1 − *A*_co_/*A* = 1 − δ_co_. Furthermore, the area of the sputtered region of the internal plate *A*_t_1__ in this design is determined by the equality *A*_t_1__ = *A*_co_, and its relative value is *A*_t_1__/*A* = *A*_co_/*A* = δ_co_.The temperatures of the surfaces *T_i_*, *i* = t_1_, t_2_, s, w are different. The temperature of the gas environment *T*_0_ is equal to the wall temperature *T*_w_. The internal plate works in cold mode, and the external one in hot mode.Two reactions can take place on each surface:
(96)$$M_{1}+\frac{n_{1}}{2m_{1}}X_{2}\overset{k_{1}(T_{i})}{\to }\frac{1}{m_{1}}M_{1m_{1}}X_{n_{1}},\;i=t_{1},\;s,\;w,$$
(97)$$M_{2}+\frac{n_{2}}{2m_{2}}X_{2}\overset{k_{2}(T_{i})}{\to }\frac{1}{m_{2}}M_{2m_{2}}X_{n_{2}},\;i=t_{2},\;s,\;w,$$where *m_j_* and *n_j_* (*j* = 1, 2) are the stoichiometric coefficients; *k_j_* (*T_i_*) (*j* = 1, 2) are the Arrhenius equation for the reaction rate constants, which have the dimension of the flux density by analogy with (56);The discharge current density
(98)$$j=j_{t_{1}}+j_{t_{2}},$$
where *j*_t_1__ is the component related to the internal plate, and it is determined by the fraction of the cut-out area δ_co_ = *A*_co_/*A* in the sputtered region of the external plate:(99)$$j_{t_{1}}=\delta_{\mathrm{co}}j.$$Current share *j*_t_2__ in (98) is equal to
(100)$$j_{t_{2}}=(1-\delta_{\mathrm{co}})j.$$Two processes occur on the surface of the external plate, which should be taken into account in the model:Thermal electron emission (the Richardson–Dushman Equation (86));Evaporation (the Hertz–Knudsen equation by analogy with (90)).At any moment in time:Fraction θ_t_1__ of the sputtered region of the internal plate is covered by the M_1*m*_1__X*_n_*1*__* film. The rest (1 − θ_t_1__) is pure metal M_1_;In the sputtered region of the external plate, the similar regions for M_2*m*_2__X*_n_*2*__* and M_2_ will be denoted by θ_t_2__ and (1 − θ_t_2__), respectively;On the *i*-th surface (*i* = s, w) due to reactions (96) and (97) and fluxes generated by the plates, a solid solution M_1_ + M_2_ + M_1*m*_1__X*_n_*1*__* + M_2*m*_2__X*_n_*2*__* is formed. On each surface, the solid solution is represented in the form of four regions with relative areas θ*_i_*1*__*, θ*_i_*1*__*, θ*_i*n*_*1*__*, and θ*_i*n*_*2*__* containing the corresponding components of the solution. In this case, it is obvious that θ*_i_*1*__* + θ*_i_*2*__* +θ*_i*n*_*1*__* + θ*_i*n*_*2*__* = 1.Each *i*-th surface (*i* = t_1_, t_2_, s, w) consumes the reactive gas to support reactions (96) and (97). Let *Q_ij_* denote the fluxes incident on the *i*-th surface (*i* = t_1_, t_2_, s, w) and participating in the formation of the *j*-th compound (*j* = 1, 2).

The independent variables in this problem are the flow of reactive gas introduced into the chamber *Q*_0_, the discharge current density *j*, and the relative area of cut-outs in the external plate δ_co_. The main dependent variable, as in all previous models, is the partial pressure *p* of the gas X_2_. Next, we present an analytical description of the physical model as a system of equations.

The system of equations includes two groups:The first of them contains equations describing the kinetics of processes occurring on all surfaces specified in the model;The second group consists of equations describing reactive gas flows to all surfaces and equations for pumping and balance of gas flows.

On each surface, the formation of M*_jm_j__*X*_n_j__* (*j* = 1, 2) and their removal proceed according to different schemes. Removal of the film from the internal plate occurs only by sputtering. On the external plate, the film removal process enhances evaporation. The steady state of the plates is described by equations like (91), in which θ_t_1__ and θ_t_2__ are unknown. These equations are independent and can be easily solved analytically:(101)$$\theta_{t_{1}}=f_{t_{1}}[k_{1}(T_{t_{1}}),\;\;\theta_{{0t}_{1}},\;\;J_{{\mathrm{tC}}_{1}}],$$
(102)$$\theta_{t_{2}}=f_{t_{2}}[k_{2}(T_{t_{2}}),\;\;\theta_{{0t}_{2}},\;\;J_{{\mathrm{tC}}_{2}}].$$

Similar values of θ*_i*n*_*1*__* and θ*_i*n*_*2*__* are used in the steady state equations of the *i*-th (*i* = s, w) surface of type (93). These equations are easily solved for unknowns:(103)$$\;\;\;\;\theta_{i_{1}}=f_{in_{1}}[k_{1}(T_{i}),\;{\;\theta }_{0i_{1}},\;\theta_{i_{1}},\;A,\;A_{\mathrm{co}},\;A_{s},\;A_{w},\;J_{{\mathrm{tM}}_{1}},\;J_{{\mathrm{tM}}_{2}},\;J_{{\mathrm{tC}}_{1}},\;J_{{\mathrm{tC}}_{2}},\;\theta_{t_{1}},\;\theta_{t_{2}}],$$
(104)$$\;\;\;\;\theta_{i_{2}}=f_{in_{2}}[k_{2}(T_{i}),\;{\;\theta }_{0i_{2}},\;\theta_{i_{2}},\;A,\;A_{\mathrm{co}},\;A_{s},\;A_{w},\;J_{{\mathrm{tM}}_{1}},\;J_{{\mathrm{tM}}_{2}},\;J_{{\mathrm{tC}}_{1}},\;J_{{\mathrm{tC}}_{2}},\;\theta_{t_{1}},\;\theta_{t_{2}}].$$

In (101), (103), (104), the fluxes of the compound M_1*m*_1__X*_n_*1*__* and metal M_1_ with densities *J*_tC_1__ and *J*_tM_1__, respectively, are described by Equations (92) and (94) for the case with a single cold target (without the second term). In (102)–(104), the fluxes of the compound M_2*m*_2__X*_n_*2*__* and metal M_2_ with densities *J*_tC_2__ and *J*_tM_2__, respectively, are described by Equations (92) and (94) in full form.

As follows from (103) and (104), the values of θ*_i_*1*__* and θ*_*i*_2__* are related to (101) and (102) for the values of θ_t_1__ and θ_t_2__. Equations (101)–(104) allow you to find all six fluxes, which are calculated by analogy with (78). In addition, the pumped-out *Q*_p_ and the total flow *Q*_0_ should be specified in the form (77) and (80), respectively.

Thus, the physicochemical model of reactive sputtering of a sandwich target is described by a system of 14 algebraic equations. Six of them set the steady state of the surfaces of two plates, the substrate, and the wall. The remaining eight set the reactive gas flows in the vacuum chamber. More details about this system can be found in [194].

The described model was used to study the process of film deposition of a binary solid solution of TiO_2_ and Ta_2_O_5_ oxides. Its chemical formula was written as Ti*_x_*Ta_1–*x*_O. These films are of interest in many areas of technology [195,196,197,198,199]. In this case, the sputtering unit consists of an internal titanium plate and an external tantalum plate.

In this problem, reactions (96) and (97) take the form
(105)$$\mathrm{Ti}+O_{2}\overset{k_{1}(T_{i})}{\to }{\mathrm{TiO}}_{2},\;i=t_{1},\;s,\;w,$$
(106)$$\mathrm{Ta}+\frac{5}{4}O_{2}\overset{k_{2}(T_{i})}{\to }\frac{1}{2}{\mathrm{Ta}}_{2}O_{5},\;i=t_{2},\;s,\;w,$$

When performing calculations on the optical spectra of the discharge, the dependence of the effective temperature $T_{t_{2}}(j_{t_{2}}^{+})$ on the ion current density of the external plate $j_{+t_{2}}$ was determined [200]:(107)$$T_{t_{2}}(j_{+t_{2}})=293+1520\left(1-e^{-0.00439j_{+t_{2}}}\right).$$

The results of calculations that were performed with the values of the parameters from Table 1, Table 2 and Table 4 are shown in Figure 6 of [194].

The dependences *p* = *f*(*Q*_0_) in [194] are given on a linear and logarithmic scale, which made it possible to reveal the influence of the internal titanium plate on the process. This effect is observed in the sections of the curves in the region of low gas incoming flow *Q*_0_ = 10^−3^–10^−1^ sccm, where the internal plate makes the transition from the metallic to the oxide mode. The change in the operating mode of both plates at different values of *Q*_0_ is associated with a different rate of removal of oxide films from their surfaces. Table 1 and Table 2 show that *S*_Ta_2_O_5__ > *S*_TiO_2__. Second, it can be shown that, at the discharge current density of more than 700 A/m^2^, the sputtering of the Ta_2_O_5_ film is supplemented by its evaporation. Therefore, the change in the steady state of the surface of the internal plate occurs at a lower rate of the chemical reaction proceeding on it and, hence, at a lower incoming oxygen flow.

The proposed model made it possible to calculate the dependence of the stoichiometric coefficient $x={\left.f(Q_{0},\;\delta_{\mathrm{co}})\right|}_{j=100{A/m}^{2}}$ in the formula Ti*_x_*Ta_1–*x*_O (see Figure 12 of [194]).

In [194], it is shown that there is an area *Q*_0_ with a negative derivative and a complete transition of the process to the steady mode with a constant value of *x*.

Concluding the review of works devoted to the modeling of reactive sputtering based on chemical reactions occurring under nonisothermal conditions, we note the main thing in this class of models.

Let us start with the disadvantage. The mathematical description of the model is complicated for application by the presence of three unknown parameters in the description of the kinetics of the chemical reaction. Moreover, if the value of the adsorption coefficient can be taken equal to unity, then the parameters of the chemical reaction constant can be determined only from experimental results by solving an optimization problem.

An advantage of the nonisothermal physicochemical model is a more correct description of the process of reactive sputtering and, as a result, the adequacy of the model, which does not require the involvement of additional mechanisms occurring on the target surface.

## 6. Summary

The study of articles published over the past 50 years on reactive sputtering and modeling of this process allowed us to identify the main stages and directions of the research development.

Reactive sputtering is a well-known technological method, the most common in various forms of magnetron sputtering. It is used for the deposition of films of simple metal compounds (nitrides, oxides, carbides, etc.) and their diverse double, triple, etc. solid solutions.

In early experimental works, a significant influence of the reactive gas concentration in the gas mixture or its partial pressure on the film growth rate, discharge voltage, and film composition was found. In addition, nonlinear effects were discovered, and it was found that with an increase in the partial pressure of the reactive gas, there is a critical value. At this value, during the sputtering process, an abrupt change in the film deposition rate was observed, in some cases, by an order of magnitude.

In later works, it was found that the partial pressure of the reactive gas was not an independent variable. It reflected only the state of the process with given values of other parameters that could be changed independently. The main ones were the incoming reactive gas flow and the discharge current (or the power released on the target). In a large number of experiments, when changing the flow or discharge current (power), the hysteresis was discovered.

For a more detailed study of reactive sputtering, many experts worked on the development of their models. All the variety of known models for reactive sputtering was actually based on two assumptions: there are two processes competing on the excited target surface (the formation of a thin layer of metal compound with reactive gas and its removal by accelerated argon ions); on the substrate and walls of the vacuum chamber, the sputtered target material is deposited, and the reactive gas molecules are chemisorbed.

At the initial stage, when constructing models, which are called “specific” in this work, the partial pressure of the reactive gas was taken as an independent variable, and only the processes occurring on the target were considered without taking into account its temperature. The difference between the models of different authors was not fundamental. The main unifying element of all cited publications was the kinetic equation for the target surface.

Further, specific models of reactive sputtering were developed. They included the surfaces of the vacuum chamber wall and the substrate. Moreover, this entailed a complication of the analytical description, in which equations for gas flows to each surface and an equation for the balance of gas flows appeared.

Specific models based on chemisorption received the most consistent development in the work of Professor S. Berg and co-workers. These models eventually became known as the Berg model, and in this work, they are called “general”. In the initial Berg model, he described the simplest process of reactive sputtering of a single metal target in a mixture containing one reactive gas. Subsequently, models of more complex processes were proposed. They included, for example, sputtering of a single target in a mixture containing two reactive gases, two targets in a mixture containing one reactive gas, etc. In the latter version, the authors included implantation and the knock-on effect in the model, excluding sputtering of the compound from the target surface in molecular form, which changed the kinetic equations for all surfaces.

Ease of study and use led to great interest in the Berg isothermal chemisorption model over the last twenty years. Specialists used it to describe practical problems. Some of them offered options for its development. In the series of publications on modeling of reactive sputtering, there were works describing other models based, for example, on the laws of thermodynamics, statistics, etc.

The second group of models was proposed by the group of Professor D. Depla. This team has consistently studied reactive sputtering and its modeling over the past twenty years.

A new reactive sputtering model called the RSD model, has been introduced stepwise. In the first step, the authors took into account only the process of implantation of reactive gas ions, which is accompanied by a bulk chemical reaction and sputtering. In this problem, the main one was the kinetic equation describing the change in the state of the target surface. In the next step, the RSD model was fully formed by including chemisorption and the knock-on effect. Subsequently, the authors aimed to improve the computational procedures for solving problems of reactive sputtering modeling. Thus, the RSD2009 and RSD2013 models, etc., appeared.

In fact, the modified Berg model has become an analog of the RSD model in its full form, including implantation and knock-on effect. The exception was the assumption about the features of compound sputtering from the target surface. In the RSD model, it took the molecular form. This greatly distinguished it from the Berg extended model.

Finally, the models developed by our scientific group formed the third approach to the problem. These models differ from all previous ones as they replace chemisorption with surface chemical reactions and remove restrictions on surface temperatures inside the vacuum chamber. Our model was called “nonisothermal physicochemical” or the Barybin model. The initial model described sputtering of a single metal target in a mixture with one reactive gas. The chemical reaction was represented using the Langmuir monomolecular adsorption isotherm and the law of mass action. In subsequent works, the Barybin model was extended to describe more complex sputtering processes in an environment containing two reactive gases, sputtering of a single hot target, and a sandwich target. Here it is necessary to pay attention to the fact that chemisorption models cannot be applied to the last two versions of the magnetron.

Concluding the article, I would like to express the hope that the modeling of reactive sputtering is far from complete. There are complex processes involving more than two magnetrons. The sandwich target sputtering unit may contain more than two plates with variations in their chemical composition. The hot target of the magnetron can operate in a liquid state. There are many practical problems of this kind. To solve them with the help of chemisorption models, first of all, it is necessary to abandon isothermality. In addition, none of the currently known models has studied the effect of diffusion of reactive gas atoms into the target. This process can become important, especially in hot targets. The possibility of including polymolecular adsorption in the model is also not ruled out.

## Figures and Tables

**Figure 1 materials-16-03258-f001:**
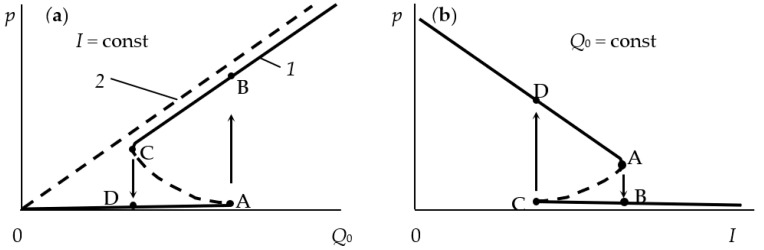
Qualitative view of the dependence of the reactive gas partial pressure during reactive sputtering on: (**a**)*—*its incoming flow; (**b**)*—*discharge current.

**Figure 2 materials-16-03258-f002:**
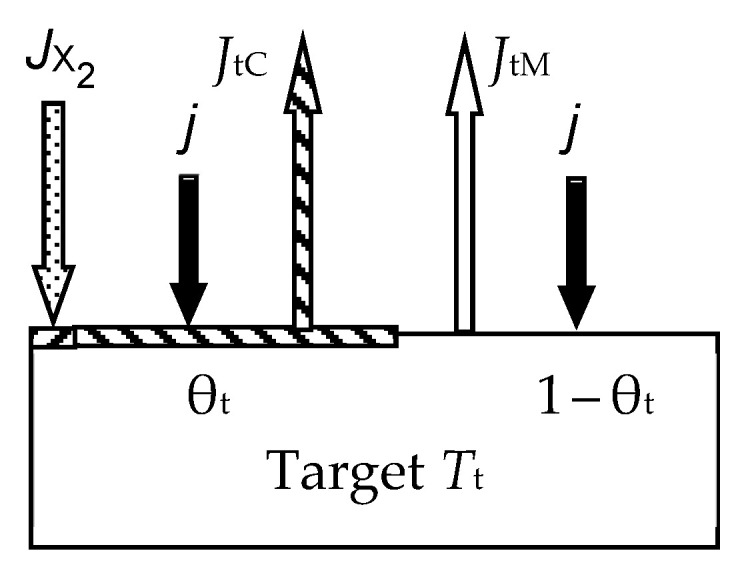
The flux densities on the target surface. Notations: *J*_X2_ is the flux density of X_2_ molecules; *J*_tM_ is the flux density of sputtered metal atoms; *J*_tC_ is the flux density of sputtered M*_m_*X*_n_* molecules; *j* is the discharge current density; θ_t_ is the fraction of the target surface covered by the compound film.

**Figure 3 materials-16-03258-f003:**
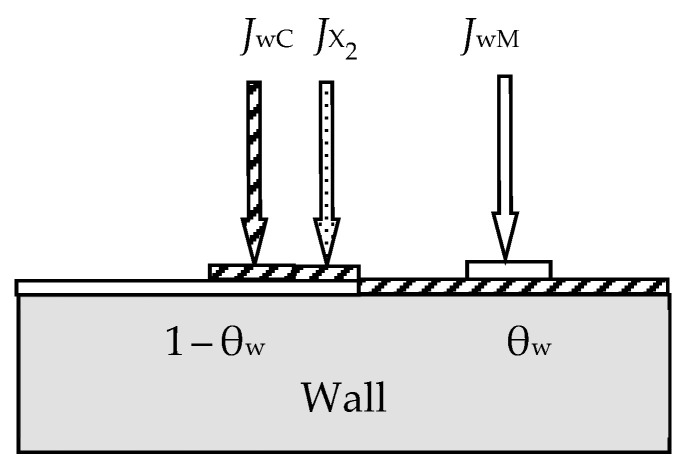
The flux densities on the wall surface. Notations: *J*_wM_ is the flux density of metal atoms; *J*_w_ is the flux density of M*_m_*X*_n_* molecules; θ_w_ is the fraction of the wall surface covered by the compound film.

**Figure 4 materials-16-03258-f004:**
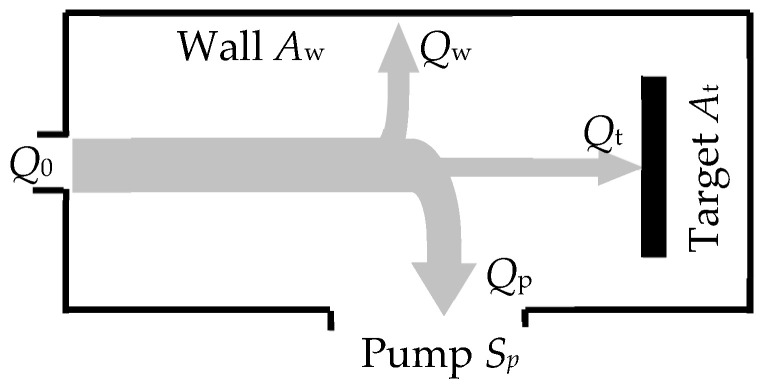
Flows on all surfaces.

**Figure 5 materials-16-03258-f005:**
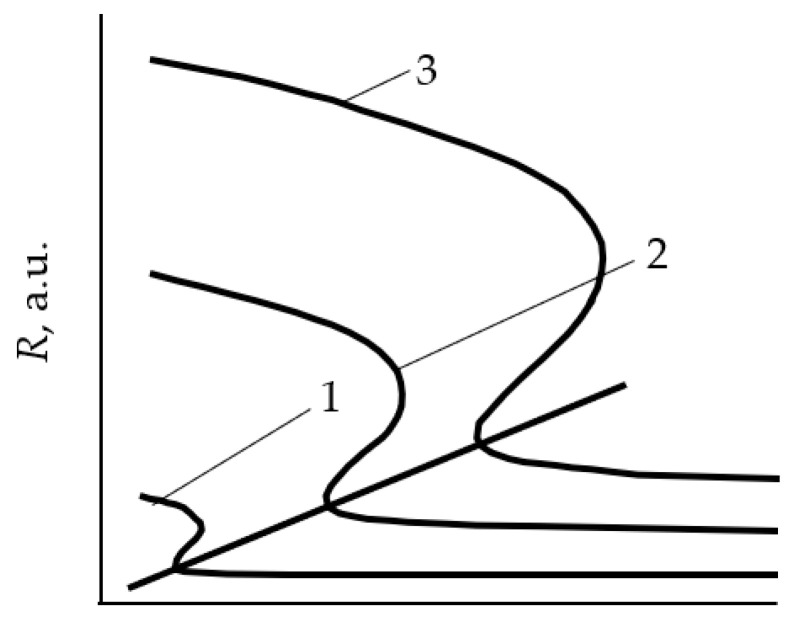
The target sputtering rate at current densities: 1—*j*; 2—3*j*; 3—5*j*/ Notations: *j* is the discharge current density.

**Figure 6 materials-16-03258-f006:**
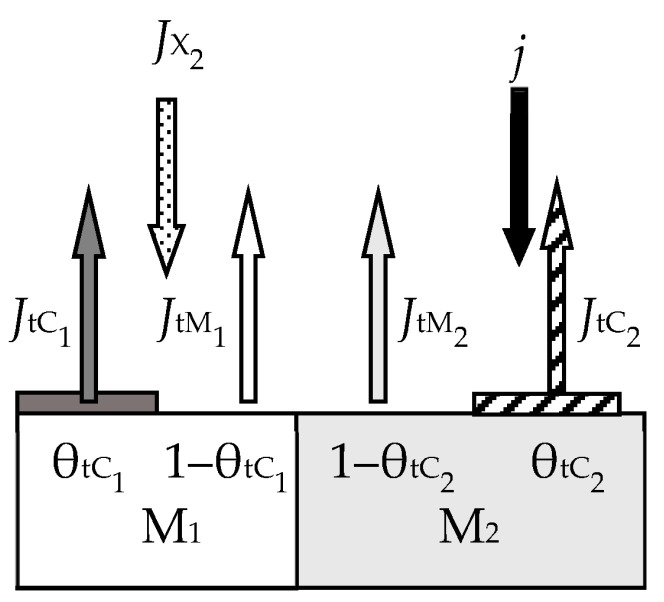
The fluxes on the target surface. Notations: *J*_tM_1__ is the flux density of sputtered metal atoms M_1_; *J*_tM_2__ is the flux density of sputtered metal atoms M_2_; *J*_tC_1__ is the flux density of sputtered M_1_X molecules; *J*_tC_2__ is the flux density of sputtered M_2_X molecules; θ_tC_1__ is the fraction of the target surface covered by the compound film M_1_X; θ_tC_2__ is the fraction of the target surface covered by the compound film M_2_X.

**Figure 7 materials-16-03258-f007:**
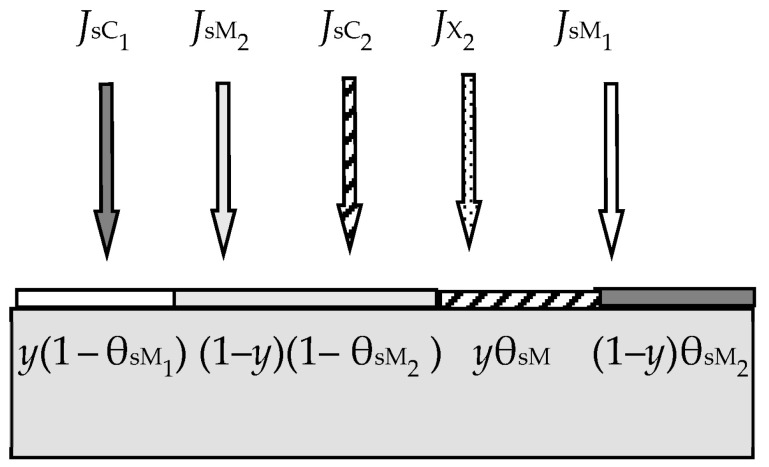
The fluxes on the substrate surface. Notations: *J*_sM_1__ is the flux density of metal atoms M_1_; *J*_sM_2__ is the flux density of metal atoms M_2_; *J*_sC_1__ is the flux density of M_1_X molecules; *J*_sC2_ is the flux density of M_2_X molecules; θ_sM_1__ is the fraction of the substrate surface covered by the metal film M_1_; θ_sM_2__ is the fraction of the substrate surface covered by the metal film M_2_; θ_sC_1__ is the fraction of the substrate surface covered by the compound film M_1_X; θ_sC2_ is the fraction of the substrate surface covered by the compound film M_2_X.

**Figure 8 materials-16-03258-f008:**
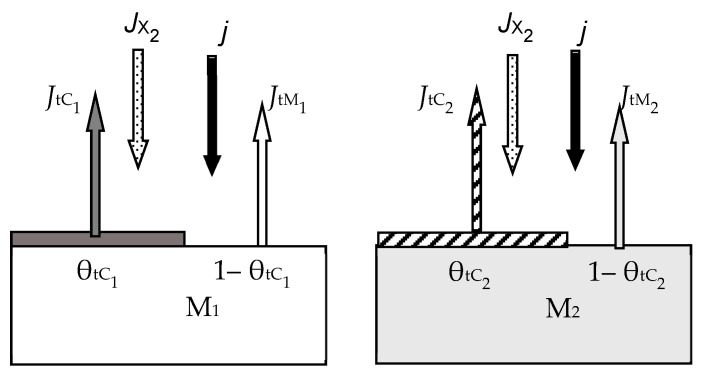
The fluxes on the surface of the targets.

**Figure 9 materials-16-03258-f009:**
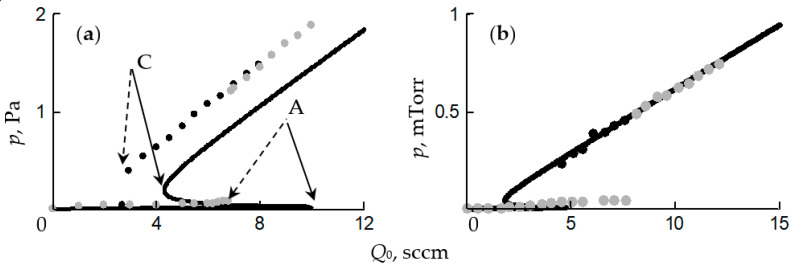
The results of the isothermal model application: (**a**)—Ta_2_O_5_; (**b**)—TiO_2_. The results of the experiment are shown by dots: gray—increase in *Q*_0_; black—decrease in *Q*_0_.

**Figure 10 materials-16-03258-f010:**
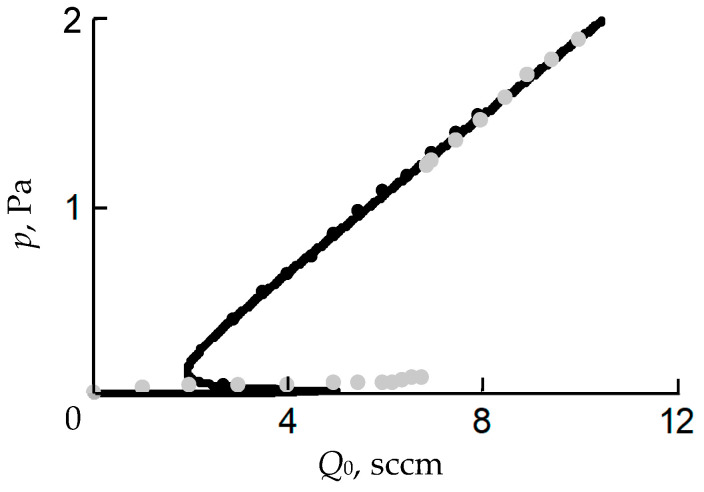
The result of applying the isothermal model for Ta_2_O_5_ at *A*_t_ = 35 cm^2^ and *S*_p_ = 8 L/s.

**Figure 11 materials-16-03258-f011:**
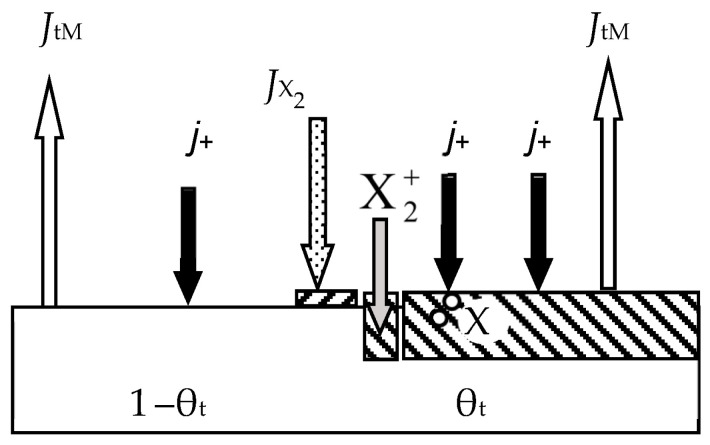
The fluxes on the target surface.

**Figure 12 materials-16-03258-f012:**
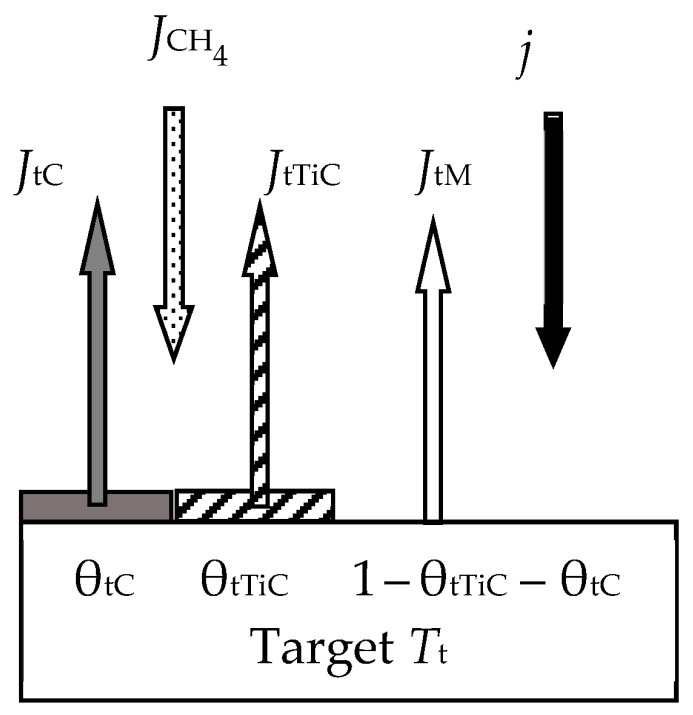
The fluxes on the target surface.

**Figure 13 materials-16-03258-f013:**
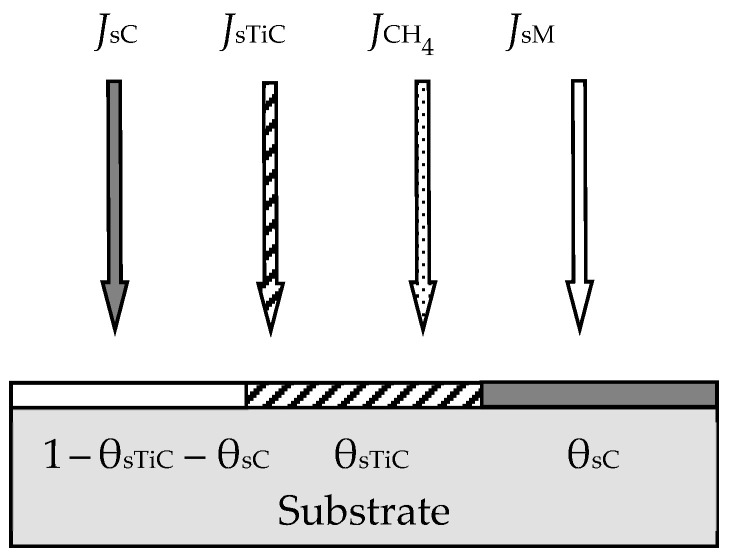
The fluxes on the substrate surface.

**Figure 14 materials-16-03258-f014:**
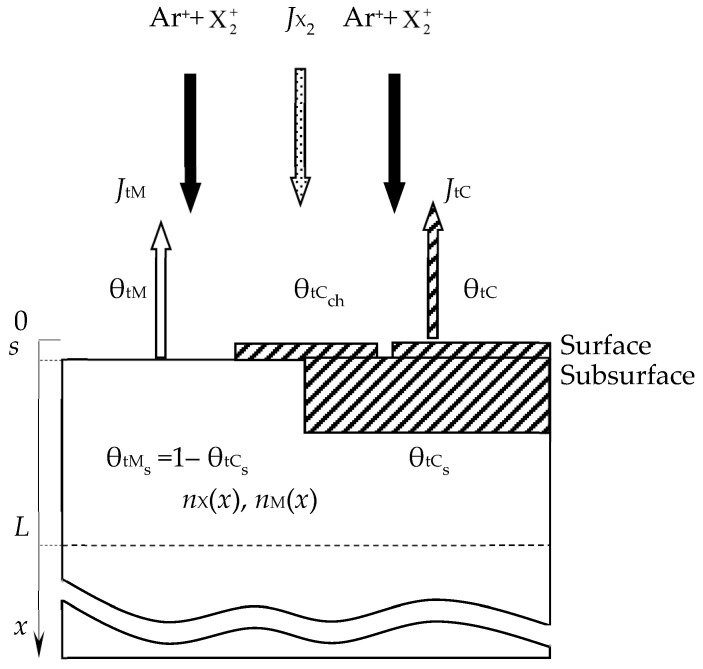
Model of target. Notations: *J*_X2_ is the flux density of X_2_ molecules; *J*_tM_ is the flux density of sputtered metal atoms; *J*_tC_ is the flux density of sputtered M*_m_*X*_n_* molecules; θ_tM_ is the fraction of the surface free from the film of the M*_m_*X*_n_* compound; θ_tC_ is the fraction of the surface occupied by the M*_m_*X*_n_* compound formed due to the bulk chemical reaction with the participation of implanted X atoms; θ_tCch_ is the fraction of the surface occupied by the M*_m_*X*_n_* compound formed due to chemisorption of X atoms; θ_tCs_ is the fraction of the subsurface region in the *s* plane occupied by the M*_m_*X*_n_* compound formed due to the bulk chemical reaction *m*M + *n*X ↔ M*_m_*X*_n_*; θ_tMs_ = 1 − θ_tCs_ is the fraction of the subsurface region occupied by free M metal atoms; *n*_X_(*x*), *n*_M_(*x*) is the volume concentration of free X and M atoms, respectively, which did not take part in the chemical reaction.

**Figure 15 materials-16-03258-f015:**
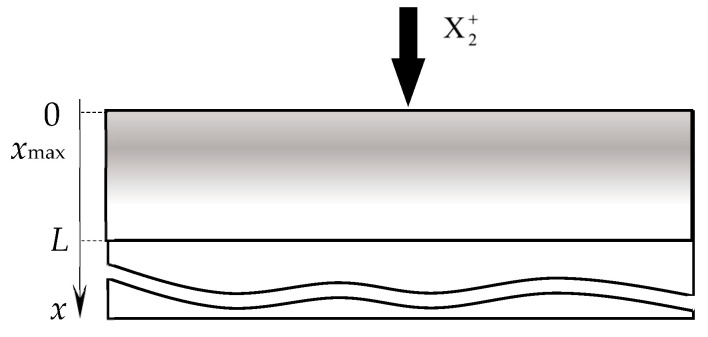
Distribution of implanted atoms in the target.

**Figure 16 materials-16-03258-f016:**
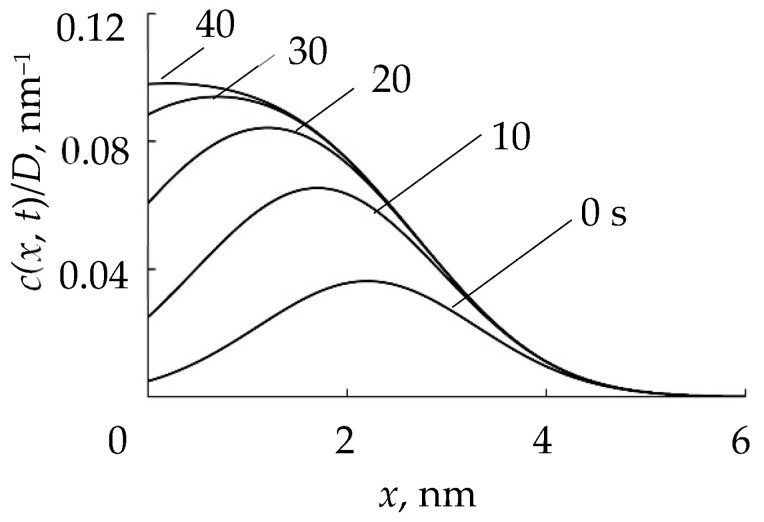
Distribution of the relative concentration of implanted ions $N_{2}^{+}$ in silicon at energies 0.45 keV and different points in time.

**Figure 17 materials-16-03258-f017:**
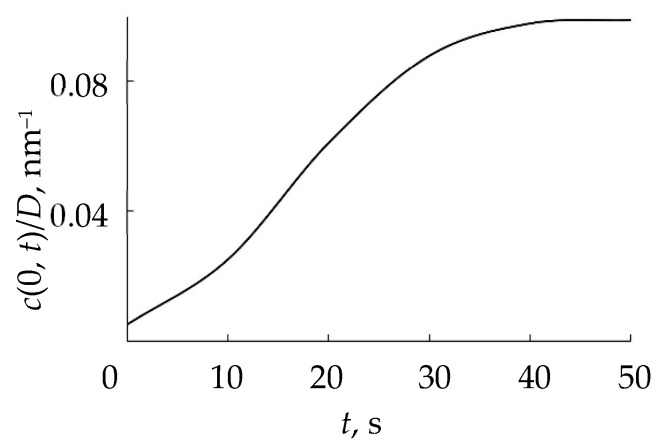
Change in the relative surface concentration of implanted ions $N_{2}^{+}$ into silicon at an energy of 0.45 keV.

**Figure 18 materials-16-03258-f018:**
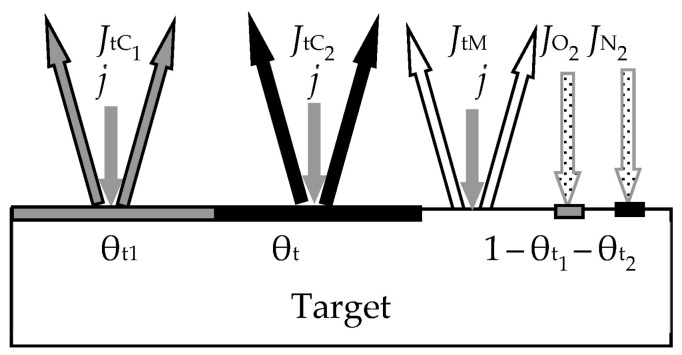
Fluxes on the target.

**Figure 19 materials-16-03258-f019:**
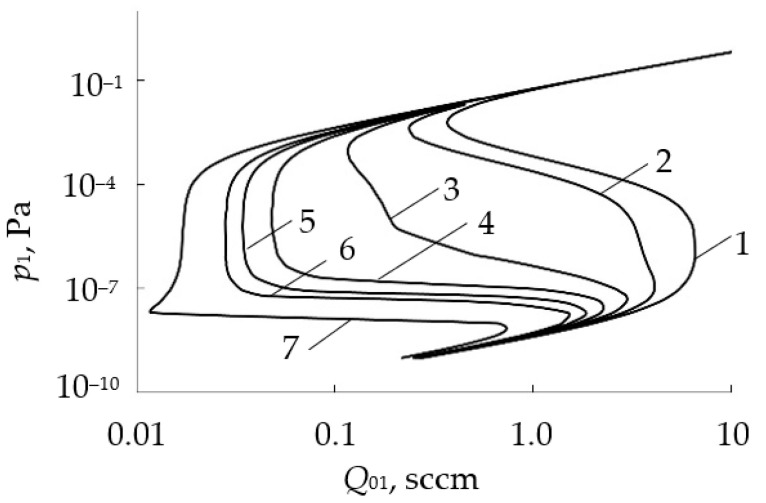
Dependences of the oxygen partial pressure on its incoming flow during reactive sputtering of a titanium target in Ar + O_2_ + N_2_ with the current density 200 A/m^2^ and an incoming nitrogen flow (sccm): 1—0; 2—0.2; 3—0.4; 4—0.6; 5—0.8; 6—1.0; 7—2.0.

**Figure 20 materials-16-03258-f020:**
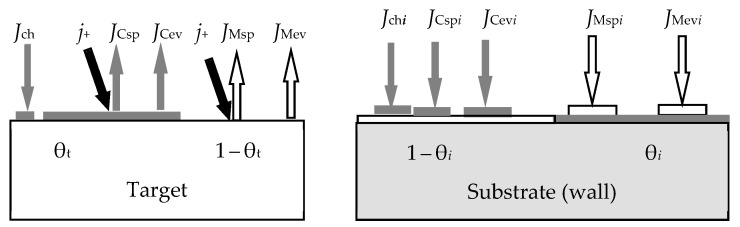
Processes on the surfaces of the target and substrate (wall) during reactive sputtering.

**Figure 21 materials-16-03258-f021:**
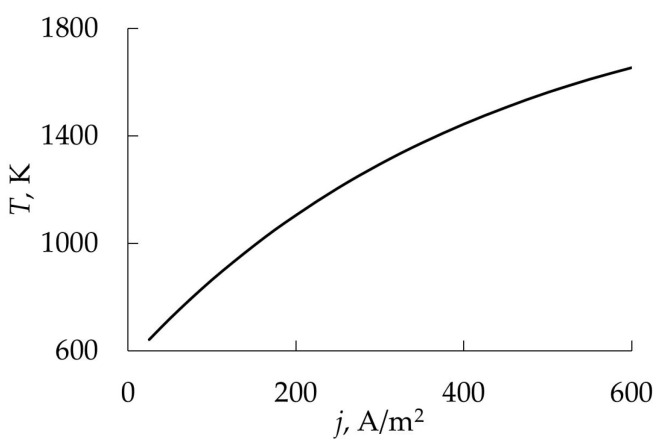
Dependence of target temperature on current density.

**Figure 22 materials-16-03258-f022:**
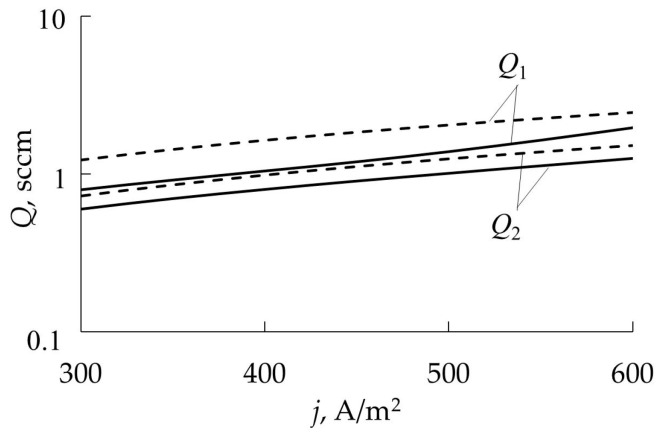
Dependences of the critical values of nitrogen incoming flow on the current density during the transition of the target: *Q*_01_—from the metallic mode to the nitride one; *Q*_02_—from the nitride mode to the metallic one (solid lines—hot target, dashed lines—cold target).

**Figure 23 materials-16-03258-f023:**
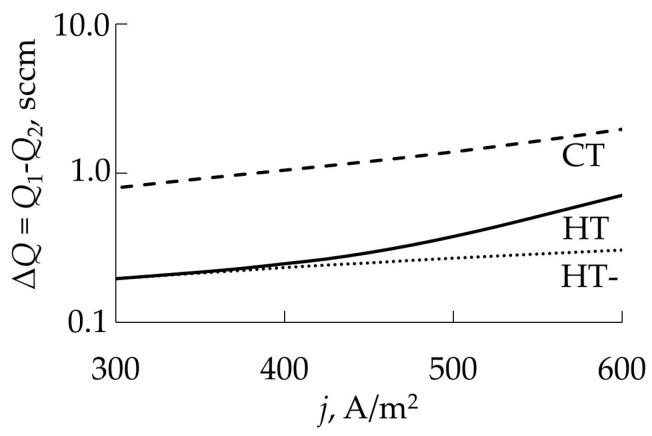
Dependences of the hysteresis width on the current density during sputtering: CT—cold target; HT—hot target; HT^−^—hot target without evaporation.

**Figure 24 materials-16-03258-f024:**
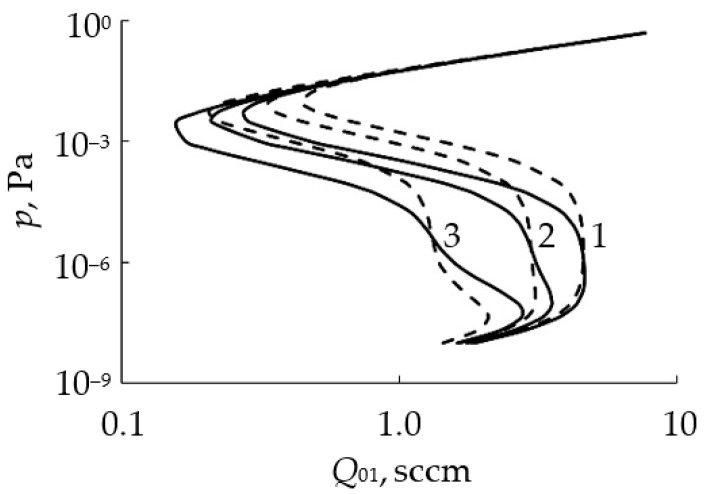
Dependence of oxygen partial pressure on *Q*_01_ for cold (solid lines) and hot (dashed lines) targets at the discharge current density of 200 A/m^2^ and values *Q*_02_ (sccm): 1—0.1; 2—0.2; 3—0.3.

**Table 1 materials-16-03258-t001:** The parameters of the model for reactive sputtering of a cold tantalum target in Ar + O_2_.

Parameter	*α* _0_	*T*_t_, K	*T*_s_, K	*T*_w_, K	*Q*_ph_, cal/mol	*N*_ph_, 10^18^ m^−2^
Value	1.0	700	600	300	10,000	14.0
Parameter	*S* _M_	*S* _C_	*S*_p_, m^3^/s	*A*_t_ m^2^	*A*_w_, m^2^	*A*_s_, m^2^
Value	0.6 [45]	0.024 [45]	0.0086	0.002	0.029	0.001
Parameter	*E_a_*1*__*, 10^−20^ J	*k*_0_1__, 10^33^ m^−2^s^−1^	*m*_O_2__, 10^−26^ kg	*m*_2_, 10^−26^ kg	*m*_n_2__, 10^−26^ kg	
Value	7.4	1.1	5.32	30	76.7	

Note. The value of *T*_t_ = 700 K is an estimate. The values of *T*_s_ and *T*_w_ are typical for real processes.

**Table 2 materials-16-03258-t002:** Parameters of the reactive sputtering process of a cold titanium target in Ar + O_2_ + N_2_.

Parameter	*α* _0_	*T*_t_, K	*T*_s_, K	*T*_w_, K	*Q*_ph_1__, cal/mol	*N*_ph_, 10^18^m^−2^	*S* _M_	*S* _C_1__	*S* _C_2__
Value	1.0	750	600	300	10,000	14.7	0.3 [175]	0.016 [176]	0.07 [165]
Parameter	*S*_p_, m^3^/s	*A*_t_ m^2^	*A*_w_, m^2^	*A*_s_, m^2^	*E_a_*1*__*, 10^−20^ J	*k*_0_1__, 10^31^ m^−2^s^−1^	*E_a_*2*__*, 10^−20^ J	*k*_0_2__, 10^29^ m^−2^s^−1^	
Value	0.025	0.00365	1.0	0.001	2.62	1.9	6.5	1.4	

**Table 3 materials-16-03258-t003:** Parameters of the model for reactive sputtering of a hot titanium target in Ar + N_2_.

Parameter	*S* _M_	*S* _C_	φ_M_, eV	φ_C_, eV	γ_M_	γ_C_	A_M_
Value	0.30 [175]	0.07 [176]	4.5 [184]	4.6 [185]	0.053 [186]	0.049 [186]	10.4 [187]
Parameter	B_M_	A_C_,	B_C_	*α* _0_	*Q*_ph_, cal/mol	*N*_ch_, 10^18^m^−2^	*k*_0_, 10^29^ m^−2^s^−1^
Value	23230 [187]	13.3 [188]	28840 [189]	1.0 [35]	10,000	14.7	1.4
Parameter	*E_a_*, 10^−20^ J	*S*_p_, m^3^/s	*A*_t_ m^2^	*A*_w_, m^2^	*A*_s_, m^2^	*T*_w_ K	*T*_s_, K
Value	6.5	0.025	0.00365	1.0	0.0003	300	600

**Table 4 materials-16-03258-t004:** Additional parameters of the model for reactive magnetron sputtering of a hot tantalum target in Ar + O_2_.

Parameter	γ_2_	γ_n_2__	φ_2_, eV	φ_n_2__, eV	*T* _t_2__
Value	0.13 [186]	0.057 [186]	4.5 [201]	5.15 [202]	$f(j_{t_{2}}^{+})$
Parameter	A_2_	B_2_, K	A_n_2__	B_n_2__, K	
Value	9.2 [187]	17260 [187]	15.3 [203]	20,000 [203]	

## Data Availability

Data is contained within the article.

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
