# Peer review of "Modeling of Reactive Sputtering—History and Development"

_materials, 2023, doi:10.3390/ma16083258_

Round 1

Reviewer 1 Report

This paper focuses on the evolution of reactive sputtering modelling and summarizes the main features of the deposition of simple metal compound films. The abstract and conclusion outlined the work well; the manuscript was logically coherent. Then, this study can guide subsequent research on reactive sputtering modelling. This will be of interest to the readership of the Journal of Materials. There are a few issues I would like to have clarified, however.

1. The formatting of many figures in the article needs attention. The names of most figures need to be more apparent. Therefore, the figure needs to be reformatted to see all the notes, such as fig 3, fig 6, fig 10-11, and so on. This is a severe format problem.

2. The name of the figure font should be the same size. For example, the legend font sizes of fig 6 and fig 7 differ.

3. The format of all formulas in the paper should be consistent.

4. It is mentioned in section 2 that the kinetic dependences presented have inflexion points during the deposition of a TiO2 film. This contradicts all known experimental results, so the author should explain why.

5. In fig 9(b), the abscissa 0 of the origin is not at its proper position.

6. In an article, the positions of a and b in fig 9 and fig 28 should be consistent.

7. Most of the models in this paper are theoretical. Some existing experimental literature can be introduced to testify.

8. This paper is mainly a review of the theoretical model, which can summarize some practical applications of the reactive sputtering model introduced in this paper.

9. Although this is a review, ending the article with some of the author's opinions and the future challenges facing technology would be nice.

Reviewer 2 Report

In this work an extensive review about the modeling process for reactive magnetron sputtering is presented. The emphasis is mainly focussed in a metallic target when forming a nitride/oxide compound, based at first on the Berg model, involving one magnetron. The subject is of current interest, since magnetron sputtering is a widely used techique for both academic/industrial purposes. I have no further suggestions, so this work can be already publised.

Author Response

Dear Colleague,

Thank you very much for your positive review of the manuscript.

Reviewer 3 Report

The paper presents a review of reactive sputter deposition processes. It appears a bit lengthy and, if possible, could be shortened (optional). There are other shortcomings which must be removed prior to publication. 

 Page 1: check page number in contents and in paper. In my opinion, page numbers are different.

Page 3, line 1: “transition” metals? What about Al, Se, In, Sn, Sb, Bi, Ba, Ca, etc., films. Delete “transition”.

Page 3, line 2: what are “their solid solutions”? Delete?

Page 3, second paragraph: “united” or “summarized”?

Page 3, second paragraph: “X2 ? What about reactive gases like H2S and H2Se for deposition of, e.g., sulphides (Fe2S) ?

Page 3, second paragraph: produces a film of “stoichiometric” composition. Maybe, but in many cases deposited films are non-stoichiometric.  

Page 3, last paragraph and page 4, first paragraph: The ion-induced electron emission is low in metallic and high in reactive mode? Is that so? In my opinion, it depends on the secondary electron emission coefficient which can be as well larger or smaller in metallic compared to reactive mode. Check with Depla et al, Ref. 186, Figs. 9 and 10.  

Fig. 1: its incoming “flow”. In my opinion, it should be “flow rate”.

All figures. Captions must be improved. For examples, a caption like “flux densities on the surface” (in figures 2, 3, 6, 7, 8, ….) is insufficient. The fluxes J must be mentioned in the caption, e.g., Jx, JtC, and JtM, as well as the meaning of y and Theta_xx

Fig. 5: “ion” current density?

Page 27: “It turned out that in the R-HiPIMS process, the probability of observing hysteresis is much lower compared to DC magnetron sputtering.” It should be mentioned already in the introduction that hysteresis effect are imported for reactive DC magnetron sputtering but much less for reactive R-HiPIMS.  

Page 29, line 12: implanted “N” atoms which is correct. However, in Figs. 17 and 18: implanted ions “N2+” which, in my opinion, is wrong. Correct.  

I am convinced that not all references of the author (23) and of Depla’s (43) group are necessary. Suggestion: delete 50% of them.  
